



# 1 Combining low-cost, surface-based aerosol monitors with 2 size-resolved satellite data for air quality applications

**Priyanka deSouza[1†], Ralph A Kahn[2], James A Limbacher[2], Eloise A. Marais[3*], Fábio Duarte[1,4], Carlo Ratti[1]**
[1] Senseable City Lab, Massachusetts Institute of Technology, Cambridge MA, United States
[2] Earth Sciences Division, NASA Goddard Space Flight Center, Greenbelt, Maryland 20771, United States
[3*] School of Engineering and Applied Sciences, Harvard University, Cambridge, MA, United States
[4] Pontifícia Universidade Católica do Paraná, Brazil
[*] Now at: School of Physics and Astronomy, University of Leicester, Leicester, UK.
[†] *Correspondence to:* Priyanka deSouza (desouzap@mit.edu)
**Abstract:**
Poor air quality is the world's single largest environmental health risk, and air quality monitoring is crucial for
developing informed air quality policies. Efforts to monitor air pollution in different countries are uneven, largely
due to the high capital costs of reference air quality monitors (AQMs), especially for airborne particulate matter
(PM). In sub-Saharan Africa, for example, few cities operate AQM systems. It is thus important to examine the
potential of alternative monitoring approaches. Although PM measurements can be obtained from ***low-cost optical***
***particle counters (OPCs)***, data quality can be an issue.
This paper develops a new method using raw aerosol size distributions from multiple, surface-based low-cost
Optical Particle Counters (OPCs) to calibrate the Multi-angle Imaging SpectroRadiometer (MISR)
component-specific, column aerosol optical depth (AOD) data, that contain some particle-size-resolved information.
The combination allows us to derive surface aerosol concentrations for particles as small as ~0.1 μm in diameter that
MISR detects but are below the OPC detection limit of ~0.5 μm. As such, we obtain better constraints on the
near-surface particulate matter (PM) concentration, especially as the smaller particles tend to dominate urban
pollution.
We test our method using data from five low-cost OPCs deployed in the city of Nairobi, Kenya, from May 1
2016 to March 2 2017. As MISR passes over Nairobi only once in about eight days, we use the size-resolved MISR
AODs to scale the more frequent Moderate Resolution Imaging Spectrometer (MODIS)-derived AODs over our
sites. The size distribution derived from MISR and MODIS agrees well with that from the OPCs in the size range
where the data overlap (adjusted-$R^2$ ~0.80). We then calculate surface-PM concentration from the combined data.
The situation for this first demonstration of the technique had significant limitations. We thus identify factors that



will reduce the uncertainty in this approach for future experiments. Within these constraints, the approach has the
potential to greatly expand the range of cities that can afford to monitor long-term air quality trends and help inform
public policy.

**Key words:** MISR, MODIS, MAIAC, aerosol optical depth, low-cost air quality monitor, particulate matter,
Nairobi, public health

## 1 Introduction
Near-surface particulate matter (PM), airborne particles, also known as aerosol, is a major pollutant that affects air
quality, and many countries are taking measures to decrease PM levels. However, efforts to monitor air pollution in
different countries are uneven. In sub-Saharan Africa, for example, few countries operate air quality monitoring
systems, and most countries lack any air quality monitoring capabilities at all, even though the limited observations
that do exist show PM levels harmful to human health (Petkova et al., 2013). This is because air quality monitoring
equipment tends to be costly to purchase (capital costs are in the range of ~USD $100,000 -USD $200,000) and
maintain, and data processing and analysis requires additional expertise and resources (deSouza, 2017; Kumar et al.,
2015; Mead et al., 2013).

Given this context, other technologies, such as low-cost air quality sensors and satellite imagery, are being examined
as alternative means of monitoring air quality. Low-cost air quality sensors, usually costing less than $2,000
(Morawska et al., 2018), have the potential to move us from a paradigm of high-cost, highly accurate, sparse
reference air quality monitoring to low-cost, more widely available air quality monitoring networks. One of the
major drawbacks of using the lower-cost sensors is that no standards or certification criteria exist for these
instruments yet, and consequently, the quality of the data they produce is of special concern (Lewis and Edwards,
2016; US EPA, 2016).

Satellite imagery, in particular space-based aerosol datasets derived from the NASA Earth Observing System's
Moderate Resolution Imaging Spectrometer (MODIS) and Multiangle Imaging Spectro-Radiometer (MISR), have
also been used to estimate near-surface particulate matter concentrations from the retrieved total-column aerosol
optical depth (AOD), with the help of aerosol transport modelling (e.g., Liu et al., 2007; Martin, 2008; van
Donkelaar et al., 2010). The advantages of satellite technology for air quality monitoring arise from the spatially
extensive measurements over time (2000-present for MISR and MODIS), and include global coverage, instrument
calibration stability, and the low incremental cost of data acquisition.



However, the challenges of using these datasets for air-quality applications are also considerable. Among the main
challenges in using satellite-derived AOD for this application are:
(1) The low temporal frequency of measurements from polar-orbiting instruments (i.e., at most, about once daily for
MODIS, and between two and nine days for MISR, depending on latitude) compared to diurnally varying pollution
levels in many settings
(2) Inaccuracies introduced in satellite aerosol retrieval algorithms by uncertain aerosol and surface optical
properties
(3) The relatively coarse retrieval-product spatial resolution and aerosol species discrimination
(4) Inability to retrieve aerosol in the presence of cloud cover, and possible sub-pixel cloud contamination elsewhere
(Duncan et al., 2014; Martonchik et al., 2009).
(5) The relationship between satellite-derived AOD and $PM_{2.5}$ is not straightforward. AOD is the integral of
atmospheric *optical* extinction from the surface to the top of the atmosphere under ambient temperature and
humidity conditions, whereas $PM_{2.5}$ is the near-surface aerosol *mass* concentration of dry particles with diameters <
2.5 μm. The relationship depends upon the aerosol vertical distribution, hygroscopic growth factor, mass extinction
efficiency, and ambient atmospheric relative humidity profile (Gupta et al., 2006). The relationship is also time
dependent and can vary across typical satellite grid-cells (Engel-Cox et al., 2004; Hu, 2009; Lee et al., 2011).

Some recent studies that apply models to derive near-surface $PM_{2.5}$ from satellite AOD measurements combine AOD
with ground-based $PM_{2.5}$ measurements from reference air quality monitors. Many early methods derived simple
empirical relationships between $PM_{2.5}$ and AOD (Engel-Cox et al., 2004; Wang and Christopher, 2003; Zhang et al.,
2009). More advanced approaches applied chemical transport models to derive near-surface $PM_{2.5}$ from the
total-column aerosol optical depths of different aerosol components, which can be done, e.g., using model-simulated
aerosol vertical distribution and aerosol-type constraints from MISR (Friberg et al., 2018; Liu et al., 2007; Patadia,
89 2013).


Many studies have focused on continental US due to the extensive surface measurements available for model
validation (Al-Saadi et al., 2005; Liu et al., 2005; Tai et al., 2010). Gupta et al., (2006) were among the first to
examine the derivation of $PM_{2.5}$ from AOD in cities on different continents: Sydney, Delhi, Hong Kong and New
York. van Donkelaar et al., (2010) used the GEOS-Chem model to determine the scaling factors between AOD and
$PM_{2.5}$ for the entire globe. Because the AOD-$PM_{2.5}$ relationship varies by region and season, it is particularly
important to test existing models, and modify them appropriately in the data-sparse regions of the world.

To respond to this challenge, the SPARTAN network is adding numerous reference-grade surface stations in poorly
sampled areas, to evaluate and enhance satellite-derived PM results (Snider et al., 2016; Weagle et al., 2018). Given
that it is unlikely many cities will have access to reference air quality monitoring instruments due to their high cost,
it is important to start examining the fusion of data from low-cost air quality monitors with that from satellites, and





to develop insights from the combination of these measurements. This paper represents the first attempt, to the best
of our knowledge, to do so.

Part of the challenge of attempting to combine these datasets is that low-cost air quality monitors on the market are
not very reliable, and their measurements tend to be much less accurate than reference monitors (Lewis and
Edwards, 2016). Many PM monitors, termed Optical Particle Counters (OPCs), measure particle counts instead of
particulate mass, and do so reliably only for particles within certain diameter ranges. For example, assumptions
about particle density as well as the number of ultrafine particles not sampled by these instruments must be made to
convert the particle counts to $PM_{2.5}$. These assumptions introduce additional uncertainties into the results.

This paper presents a novel method linking the size-resolved information in MISR AOD component-specific
retrievals with the ground-based aerosol size distribution derived from the raw particle counts of surface-based
OPCs. As MISR passes over countries near the equator only once in about 8 days, we use monthly-MISR aerosol
climatology to scale the more frequent (twice-daily near the equator) MODIS-derived AOD.

As a first attempt at testing the method, we apply it to five Alphasense OPC-N2 low-cost monitors[1] deployed from
May 2016 to March 2017 in Nairobi, a growing metropolitan area in sub-Saharan Africa. The Nairobi case entails
some important limitations for the current application; the AOD over the region was relatively low, there were no
independent measurements of aerosol vertical distribution or any surface-based, high-quality reference air quality
monitors to help with validation. However, it is the only location where we have a significant record of coincident,
ground-based low-cost OPC data. As such, we have to make assumptions in this first demonstration of the
technique, which we detail, and mitigate to the extent possible, in this paper.

Section 2 provides an overview of the ground-based and satellite datasets involved in this study, as well as the model
simulations used to constrain the aerosol vertical distribution. Section 3 describes in detail the method we developed
for combining the surface and satellite data. Section 4 contains the results of applying this method in Nairobi. Our
conclusions appear in section 5, where we also summarize the factors that will reduce the uncertainties involved in
combining data from low cost monitors with satellite observations in future deployments.

---

[1] Alphasense OPC-N2 product page URL: http://www.alphasense.com/index.php/products/optical-particle-counter/
Last accessed 15.12.2016)



## 2 Data

### 2.1 Ground-Based measurements:

The Alphasense OPC-N2 monitor is a low-cost Optical Particle Counter, costing USD $450, that works by using focused light from a (~ 5V, 175 mA, 658 nm) laser to illuminate one aerosol particle at a time, and then measuring the intensity of light scattered. The amount of scattering is a function of the size, shape, and composition of the aerosol, and especially for spherical particles such as those most likely to dominate in the study region, the measurements can be calibrated using monodisperse particles of known size (Sousan et al., 2016). The Alphasense OPC-N2 is unique among low-cost sensors as in addition to PM estimates, it reports the raw particle counts in 16 bins based on particle diameter, ranging from 0.38 μm to 17.5 μm, which is critical to our method. The bins are tabulated in Table S1 in Supplementary Information. Sousan et al. (2016) discuss the accuracy of these count measurements in detail, and note that they agree well with reference instrument measurements for coarser particles (> 0.78 μm in diameter), but underestimate the particle counts for finer particles.

As the OPCs cannot detect particles with diameters < 0.38 μm, Alphasense provides software to extrapolate the particle counts, as needed to estimate the contribution from aerosols having diameters < 0.38 μm. The number of particles per volume of air in all bins can be obtained by dividing the particle counts of each bin by the flow rate and sampling duration. The Alphasense company proprietary data reduction algorithm makes assumptions about the particle density and volume of aerosols in each bin to calculate $PM_1$, $PM_{2.5}$ and $PM_{10}$ data from the particle count data.

Details about the Nairobi OPC deployment can be found in section S1.1 in Supplementary Information.

### 2.2 Satellite Data

Although passive remote sensing has significant limitations for air quality applications at present, it offers substantially more frequent, global-scale aerosol constraints than any other measurement technique. Starting in December 1999, the National Aeronautics and Space Administration (NASA) launched a series of Earth Observing System satellite sensors, including the two instruments we use in this experiment: the Multiangle Imaging SpectroRadiometer (MISR) on board the Terra satellite (Diner et al., 1998), and two Moderate Resolution Imaging Spectroradiometer (MODIS) sensors (e.g., Remer et al., (2005)), one each aboard the Terra and Aqua satellite platforms.



### 2.2.1 MISR Research Algorithm AOD and Particle Properties

MISR is one of five instruments aboard the Terra satellite. It measures sunlight reflected from Earth in each of nine cameras pointed at different view angles, from $+70^0$ through nadir to $-70^0$ along the satellite flight path, in each of four spectral bands (446, 558, 672 and 866nm) (Diner et al., 1998). This multi-angle design allows MISR to observe the atmosphere through effective slant path ranging from one (i.e., vertically down) to three (i.e., at steep forward and aft angles). This geometry produces scattering angles between the sun and viewing vectors ranging from approximately $60^0$ to $160^0$ in mid-latitudes. The combination of multi-spectral and multi-angular observations provides information about aerosol amount and microphysical properties, such as particle size and shape (Kahn et al., 2001; Kahn and Gaitley, 2015).

MISR algorithms retrieve aerosol properties by selecting from among the optical models for an assumed set of aerosol component mixtures. A "component" is a candidate aerosol type of specified, uniform composition and size distribution. The top-of-atmosphere reflectances simulated for each mixture are calculated and compared with the corresponding MISR observations, to determine the mixtures that fit the data within certain acceptance criteria; these are reported by the algorithm as the "successful mixtures" likely to be present (Diner et al., 2005; Limbacher and Kahn, 2014; Martonchik et al., 2009). Each mixture contains up to three individual aerosol components, where the percent contributions of all the components to the mixture mid-visible AOD sum to 100%.

The MISR Standard Aerosol retrieval algorithm uses a universe of 74 mixtures. The eight aerosol components in the MISR Standard Version 22 and 23 products are labelled: 1, 2, 3, 6, 8, 14, 19, and 21 as reported in Tables 1 and 2 in Kahn and Gaitley (2015) and reproduced in Table S3 in Supplementary Information. The components are named based on single-scattering albedo (SSA): light-absorbing or non-absorbing, particle shape: spherical, non-spherical grains or spheroids, and effective radius. Under favorable retrieval conditions (e.g., when total-column mid-visible AOD exceeds about 0.15 or 0.2), the MISR algorithm is able to distinguish between three and five bins in column-effective particle size (Kahn and Gaitley, 2015).

The spectral extinction coefficients for each aerosol component are included in the MISR Aerosol Physical and Optical Properties (APOP) file, available from the NASA Langley Research Center (LARC) Atmospheric Sciences Data Center (ASDC)[2]. The MISR Standard aerosol data product provides AOD values and success flags: (i.e., whether a mixture is an adequate fit to the observations to be considered a "successful" match) for each aerosol mixture, based on estimated measurement uncertainties.

---

[2] https://eosweb.larc.nasa.gov/sites/default/files/project/misr/DPS_v32_RevL.pdf (Last accessed on August 12, 2019)





In this paper, we use the MISR Research Aerosol retrieval algorithm (RA; Limbacher and Kahn, 2014; 2017)
applied to MISR Level 1B2 radiance data, to derive AOD estimates for the eight MISR aerosol components. The
RA can be run with different sets of aerosol components, including the 74-mixture set used in the MISR Standard
Algorithm, and reports column-effective aerosol properties at any desired spatial resolution down to the MISR pixel
resolution of 1.1 km x 1.1 km. In addition to producing results at a finer spatial resolution than the MISR Standard
aerosol product, the RA also offers significantly better MISR aerosol retrieval results for air quality and other
applications because of empirical calibration corrections(Limbacher and Kahn, 2015), better treatment of surface
boundary conditions, and other refinements (Limbacher and Kahn, 2017, 2014, 2019).

Data from MISR on its own rarely contains more detail than qualitative particle size and shape, so
particle-composition-related information that could be used to distinguish different sources or to assess particle
moisture content is lacking, except where detectable differences in other parameters, such as particle shape (e.g.,
non-spherical dust vs. spherical smoke or pollution particles) and particle light-absorption (e.g. "dirty" vs. "clean")
make these distinctions possible (Kahn et al., 2001; Kahn and Gaitley, 2015; Liu et al., 2007). MISR aerosol-type
retrieval uncertainty is assessed generally by Kahn and Gaitley (2015), and we rely on these results to indicate the
expected uncertainties here. Specifically, we enforce a lower bound of 0.15 on mid-visible AOD for accepting
MISR-retrieved particle size distributions. We assume that the aerosol components follow log-normal size
distributions, and extract the size distribution of the MISR aerosol components at diameters ranging over the MISR
size-detection range of about 0.1-3 μm.

For more details of the MISR data over the OPC-N2s in Nairobi refer to section S1.2.1 in Supplementary
Information.

### 219  2.2.2 MODIS MAIAC AOD

MODIS samples every location on the globe about twice a day, but lacks particle size information (e.g., Levy et al.,
2013). As aerosol type appears to be fairly constant on monthly timescales, we scale the MODIS-MAIAC
(MultiAngle Implementation of Atmospheric Correction) AOD retrieval product (Lyapustin et al., 2011a; Lyapustin
et al., 2011b), with available, particle-size-resolved AOD from MISR over each month.

MODIS has 36 spectral channels, designed to provide a wide variety of biogeophysical information. Unlike MISR,
which uses near-simultaneous, multiangle observations for aerosol-surface retrievals, MODIS offers single-view,
broad-swath, multi-spectral data. The MAIAC algorithm applies image-based processing techniques to analyze
MODIS time-series, i.e., multiple views of each surface location, in different parts within the MODIS swath (and
therefore different view-angles), acquired over a sliding, 16-day orbit-repeat cycle.  This non-coincident multi-angle
approach produces cloud detection, AOD and atmospheric correction over both dark vegetated land and a range of



brighter surfaces, at 1 km x 1 km resolution (Lyapustin et al., 2012). Compared to operational MODIS retrievals,
MAIAC AOD has similar accuracy over dark and vegetated surfaces, and higher accuracy over brighter surfaces
(Lyapustin et al., 2011a; Lyapustin et al., 2011b).

For details about MAIAC AOD over Nairobi during the study period, refer to Supplementary Information section
S1.2.2

### 2.3 GEOS-Chem Aerosol Vertical Scaling

GEOS-Chem simulations were used in our study to provide a constraint on the vertical distribution of the aerosols,
because AOD from the satellites is a column-integrated quantity, whereas $PM_{2.5}$ is assessed near-surface. The
GEOS-Chem model is driven with GEOS-5 assimilated meteorology from the NASA Global Modelling and
Assimilation Office (GMAO) at $0.5^0$ x $0.667^0$ horizontal resolution (Bey et al., 2001). The model is nested over the
African continent and boundary conditions are from a global simulation at $2^0$ x $2.5^0$. Natural emissions are from
MEGANv2.1 for biogenic volatile organic compounds (VOCs) (Guenther et al., 2012), for soil NOx (Hudman et al.,
2012), and for lightning $NO_x$ (Murray et al., 2012). Biogenic isoprene emissions are updated using the improved
model developed by (Marais et al., 2014). Open fire (biomass burning) emissions are from GFED4 (van der Werf et
al., 2010). Inventories of anthropogenic emissions in Africa include DICE-Africa for cars, motorcycles, traditional
biofuel use (fuelwood, charcoal, crop residue), charcoal production, ad hoc oil refining, backup generators, kerosene
use, and gas flares (Marais and Wiedinmyer, 2016). Pollution from industrial and on-grid power generation are from
EDGARv4.2 for $SO_2$, $NO_x$, and CO (EC-JRC/PBL, 2011), RETROv2 for VOCs (Schultz et al., 2007), and (Bond et
al., 2007) for black carbon (BC) and organic carbon (OC). Detailed gas and aerosol chemistry are described by (Mao
et al., 2013, 2010).

Details about the model simulations we used for the Nairobi case, as well as our attempts to validate the vertical
distribution of aerosol obtained from the GEOS-Chem model, are provided in section S1.3 in Supplementary
Information.

### 3 Methodology


Our approach uses the size distribution of the aerosol components from MISR retrievals to constrain the size
distribution derived from low-cost OPCs. The satellite size distribution data is encoded in the fractional contribution
of each MISR component AOD to the total MISR AOD. We use the 'monthly' effective fraction of each MISR
component AOD to scale the more frequent MAIAC AODs, yielding AOD values parsed out for the individual
MISR components on a more frequent basis. In particular, the constraint on the aerosol size distribution from MISR
remote-sensing data is especially important for particles with diameters < 0.54 μm, which the OPC cannot detect.



Obtaining an understanding of the size distribution between 0.1 and 0.54 μm allows for better estimation of $PM_{2.5}$
from the combined MISR and OPC measurements. We assess the assumptions required for this analysis in Section 5.
**3.1 Step 1: Estimate the ground-based size distribution of aerosols at each site from the**
**Alphasense-OPC N2 monitors**
We obtain the lognormal size distribution: dN/d(ln(d)), from the Alphasense OPC-N2 ground-based data, at the time
of the Terra overpass, for the diameter at the mid-point of each OPC bin using Equation 1.

$$\frac{dN}{dln(d)} = \frac{\Delta n}{ln(Dupper) - ln(Dlower)} \times \frac{1}{flow\ rate(\frac{ml}{s}) \times 10^{-6} \times (\frac{m^3}{ml}) \times sampling\ time}$$   (1)

Here $D_{upper}$ and $D_{lower}$ are the upper and lower diameters of each OPC bin. $\Delta n$ is the number of particle-counts in each
bin. N is the averaged number concentration of particles (units: #/volume of air) over the seven-minute Terra
overpass. The number concentration units derived from Equation 1 are #/ml. We thus multiply the result by $10^6$ to
convert the number concentration from our surface monitors to Number of particles (#) /$m^3$.

Equation 1 uses only the raw particle counts from the OPC. We do not include the first bin (0.38-0.54 μm) in this
analysis, as the error in the number concentration measurement for this bin is the highest (Sousan et al., 2016). Note
that the mode diameter of urban aerosol tends to be ~ 0.2 μm. Unfortunately, the Alphasense OPC-N2 only 'sees'
larger aerosols. This is a key reason for combining the OPC data with the satellite retrievals. In future deployments,
other instruments that can see the smaller particles can be used.
**3.2 Step 2: Estimate stable and consistent aerosol size-resolved information from satellite data**
We estimate the corresponding size distribution of surface particulate matter from MISR and MAIAC AOD
information by calculating the monthly effective near-surface AOD for each of the eight MISR aerosol components.

We denote the column fractional AOD for each aerosol component (listed in Table S3 in Supplementary
Information), as $AOD_{i,k}$: the mid-visible AOD fraction of component $i$ in the kth MISR atmospheric column
retrieval. It is calculated as the mixture-AOD-weighted AOD from all passing mixtures for component $i$ in the MISR
RA aerosol climatology.

**Step 2a.** *Estimate the near-surface fraction of the satellite AOD.* We estimate the fractional AOD for each aerosol
component residing in the lowest atmospheric layer of the GEOS-Chem model (up to ~ 130 meters above the
surface), by scaling the total-column fractional AOD with the simulated aerosol vertical profiles from GEOS-Chem
using Equation 2.




$$AOD_{N\text{-}Si} = \frac{GEOS\ Chem\ lower\ AOD}{GEOS\ Chem\ column\ AOD} \times MISR\ AODi \qquad (2)$$

Here N-S denotes Near-Surface.

**Step 2b.** *Associate the near-surface AOD with particular aerosol species in the model.* Given the difference
between the MISR aerosol components and the GEOS-Chem aerosol species, we use an approach similar to Liu et
al. (2007) to connect the two. Specifically, we sum GEOS-Chem AOD values for spherical species, SO4-NH4-NO3,
OC and BC. We then calculate the ratio of the AOD for these species in the lowest GEOS-Chem atmospheric layer
to the total columnar spherical-species AOD as the scaling factor for the MISR spherical components. For the very
large spherical (MISR aerosol component 6) and non-spherical components (MISR aerosol components 19 and 21),
we use the ratio of GEOS-Chem dust AOD in the lowest layer to the total column dust AOD (Kahn and Gaitley,
2015).   Henceforth, we refer to MISR component-specific, ***near-surface*** fractional AODs as MISR fractional
AODs.

**Step 2c.** *Derive the satellite-component size distribution contributions to specific sizes.* We now obtain the particle
properties from the MISR RA needed to constrain the OPC aerosol size distribution for sizes smaller than 0.54 μm.
Depending on retrieval conditions, if the aerosol retrieval is successful, MISR is able to distinguish aerosols in about
3-5 size bins (section 2.2.1). The MISR RA uses these data to constrain a universe of possible aerosol mixtures to a
subset of components that fit the data best. Although there is uncertainty in the details of the size distribution, the
instrument provides consistent and stable retrievals over large areas and for a long period of time. Similarly, the
process of constraining the universe of MISR aerosol types present is also consistent and stable over time. The
corresponding lognormal size distribution: dN/d(ln(d)) of all the aerosol components from the satellite data is
obtained from Equations 3 and 4a.

$$Si(d) = \frac{e^{\frac{-(ln\,(d)\,-ln\,(dci)\,)^2}{2(ln\,(\sigma i)\,)^2}}}{ln\,(\sigma i) \times \sqrt{2\pi}} \qquad (3)$$

$$\frac{dN}{dln(d)} = \sum_{i=1}^{8} N_{N-Si} \times Si(d) \qquad (4a)$$

In Equation 3, $S_i(d)$ is the normalized size distribution of MISR aerosol component i. The representative size
parameters are, specifically, the characteristic diameter ($dc_i$) and the distribution width ($\sigma_i$) for each of the eight
MISR aerosol components. Note that the upper and lower diameters of each aerosol component are considered in
this analysis. Based on the retrieval algorithm assumptions, the size distribution of an aerosol component for
diameters outside the range of each component is 0. For the Nairobi cases, only small, spherical particles and
medium-coarse particles contribute significantly to the MISR-retrieved AOD (Table 2). $N_{N-Si}$ is the total number
concentration of each MISR aerosol component present near-surface for each observation.

The size distributions Si(D) for MISR aerosol components 2, 8 and 14 are the same (Table S3). MISR aerosol
components 2, 8, and 14 represent optical analogs of typical urban pollution with different light-absorption
properties. We rewrite Equation 4a, grouping these three components into one aggregate term in Equation 4b. Here
$N_{N-S\{2,8,14\}}$ is the total near-surface number concentration of components 2, 8 and 14. The index i here runs only
over the remaining MISR aerosol components: 1,3,6,19,21.

$$\frac{dN}{dln(d)} = \sum_{i=(1,3,6,19,21)} N_{N-Si} \times Si(d) + N_{N-S\{2,8,14\}} \times S_{\{2\ or\ 8\ or\ 14\}}(d) \qquad (4b)$$


Importantly, the column-effective size distribution from Equation. 4b, derived from the MISR retrievals,
corresponds to the surface-measured value from Equation 1 only if the near-surface aerosol properties are
representative of the entire atmospheric column. Due to a lack of additional observational constraints, we must
accept this as an assumption, along with the corresponding uncertainty. The assumption will be favored in places
where the aerosol load is concentrated near-surface, which is common when the aerosol column is dominated by
local sources. This is likely the case for many urban regions and is supported by the high correlation between MISR
or MAIAC $AOD_{N-S}$ and OPC $PM_{2.5}$ in Nairobi when AOD > 0.15 (see section S2 in the Supplementary Information).
The size distribution of the total aerosol derived from a MISR retrieval is a sum of the size distributions of
individual aerosol components, as represented in Equation 4.

**Step 2d.** *Formulate the satellite constraint on size-specific surface concentration so it can be regressed against*
*the OPC data.* By definition, $AOD_{558}$ is proportional to [the number concentration of aerosols] x [the extinction area
of each particle at 558 nm wavelength] x [the path over which AOD is assessed (which here is 130 meters
vertically)]. In order to obtain near-surface number concentration of each aerosol component using this physical
definition of AOD, we assume a uniformly mixed, near-surface aerosol, with the AOD measured in all cases over a
vertical path through the first 130 m of the GEOS-Chem model. As shown in Equation 5, for each aerosol
component, a dimensionless proportionality constant multiplied by the $AOD_{N-S}$/path length (130 meters) x spectral
extinction coefficient is the number concentration of particles, summed over the path, per unit area. The spectral





extinction coefficients of each aerosol component can be found in Table S3. The near-surface number concentration
of each aerosol group is thus represented as:

For MISR aerosol components: 1,3,6,19,21:

$$N_{N-S(1,3,6,19,21)} = \Gamma_{i=(1,3,6,19,21)} \times \frac{AOD_{N-Si}}{130\ m \times 10^{-12} \times \frac{(m)^2}{(\mu m)^2}\ optical\ extinction\ coefficient\ at\ 558\ nm\ (\mu m)^2_i}$$
(5a)

For the aggregate MISR aerosol group comprising of MISR aerosol components: 2, 8 and 14:
$$N_{N-S\{2,8,14\}} = \Gamma_{\{2,8,14\}} \times \sum_{i=\{2,8,14\}} \frac{AOD_{N-Si}}{130\ m \times 10^{-12} \times \frac{(m)^2}{(\mu m)^2} \times optical\ extinction\ coefficient\ at\ 558\ nm\ (\mu m)^2_i}$$
(5b)

The spectral extinction coefficients obtained from Table S3 are in units of $(\mu m)^2$. To convert this to $m^2$, we multiply
these coefficients by $10^{-12}$. The number concentration $N_{N-Si}$ in Equations 5a and 5b has units $\#/m^3$. $\Gamma i$ is a
dimensionless scaling parameter, needed to relate the modeled aerosol number concentration of each component to
the actual number concentration present from the OPC measurements. We expect this value to be a constant, because
the MISR retrievals are stable and consistent over time. We derive this parameter using the ground-based size
distribution from the OPC-N2s, in the size range where the surface instruments have sensitivity.

**Step 2e.** *Increase the number of satellite data points by scaling MODIS AOD with MISR sizes*. To increase the
satellite dataset, we use the average fractional AOD of each MISR aerosol component for a given month over a
specific site to parse the total AOD from the more frequently sampled MAIC product, using Equation 6 to represent
the MISR component fraction, and Equation 7 to calculate the corresponding MAIAC value.
$$MISR\ AOD_{N-S,month,i} = \frac{\sum_{j=1}^{n} MISR_{N-S\ i}}{n}$$ (6)

$$MAIAC_{N-Si} = MAIAC \times \frac{MISR\ AOD_{N-S,month,i}}{\sum_{i=1}^{8} MISR\ AOD_{month,i}}$$ (7)
Here MISR $AOD_{N-S,month,i}$ is the effective MISR near-surface AOD for component *i* over a given surface site for a
specific month of the year (obtained by averaging the available data, with the assumption of negligible change in





particle properties over the month, as discussed in Section 2.2.1), and $n$ is the number of MISR $AOD_i$ retrievals for
that month. The AOD assigned to each MISR component $i$, based on scaling a given MAIAC AOD retrieval, is
denoted $MAIAC_i$, For the remaining analysis, we use the scaled $MAIAC_{N-S,i}$ instead of $MISR_{N-S,i}$ in Equations 5a and
5b.

**Step 2f.** *Regress the satellite near-surface, size-constrained particle concentration constraints against the OPC*
*data to obtain a more complete near-surface aerosol size-concentration distribution*. To appropriately link the
size-distribution from the OPCs with the MISR retrievals, we would ideally aggregate the OPC size bins in a similar
way MISR does: very small, small, medium and large, calculate the OPC size distribution at the mid-point of these
bins, and fit these size distributions with the size distribution derived from MISR. However, as the OPC has
predefined bins, we assume that for favorable retrievals, each aerosol component follows a log-normal size
distribution, consistent with the MISR algorithm assumptions. We use Equation 4 to extract the size distribution of
the total aerosol from MISR measurements that corresponds to the mid-point of each pre-existing OPC bin within its
range of sensitivity. Although the OPC counts particles for 16 diameter bins between 0.38 and 17 μm (Table S1), we
perform the OPC-MISR regression analysis only within the diameter range 0.54-2.55 μm for which both MISR and
the OPCs have adequate sensitivity. This corresponds to six of 16 OPC size bins, Bin 2-Bin 7 (Table S1). When we
use the MAIAC data, we still rely on the size information obtained from the MISR retrievals to represent aerosol
size distribution.

We perform the regression analysis, substituting the right side of Equation 1 into the left side of Equation 4b, and
substituting the right side of Equations 5a and 5b for the two $N_{N-Si}$ terms on the right side of Equation 4b. We can
then evaluate the $\Gamma i$, based on the relationship between the surface-monitor size distribution on the left side of this
equation (obtained from Equation 1), and the satellite values represented on the right side, for each coincident
observation. The $\Gamma i$ are essentially the aerosol-group-specific adjustment factors required to equate the near-surface
aerosol number concentration measured by the surface monitor with that derived from the satellite. After calculating
$\Gamma i$, we can calculate $N_{N-Si}$ using Equations 5a and 5b.
**3.3 Step 3: Calculate PM$_{2.5}$ from the number concentration of the different MISR Aerosol Groups**
In the final step, we calculate PM$_{2.5}$ using the 'OPC-calibrated' aerosol size distribution from MISR. As is already
evident from the discussion above, it is not straightforward to obtain quantitative PM$_{2.5}$ values from the particle size
distribution information derived from satellite passive remote sensing. Further, Alphasense uses a proprietary
algorithm to convert particle counts to dry mass. Particle counts in each of the 16 bins are multiplied by the volume
of particles under ambient conditions in each bin assuming spherical particle shape, an assumed particle density, and
a factor corresponding to the ISO respirable convention for PM$_{2.5}$. Assumptions are made about the efficiency of the
instrument inlet as a function of particle size, and about the size distribution functional form, to obtain the volume of



particles within each size bin. The total is then divided by the sampling time and sample flow rate to calculate the
mass obtained per unit volume of air. Given these assumptions, we have more confidence in observed *differences* in
the measurements than in the reported absolute concentration values. Our interpretation of the results in the next
section proceeds with this in mind. Assuming spherical particles, the normalized volume distribution per particle for
MISR aerosol component *i* is:
$$v_i(d) = \sum_{j=1}^{n} \frac{\pi d^3}{6} \times \frac{e^{\frac{-(\ln(d)-\ln(dcj))^2}{2(\ln(\sigma j))^2}}}{d \times \ln(\sigma j) \times \sqrt{2\pi}}$$
(8)

Note here the index *i* corresponds to MISR aerosol components: 1,3,6,19,21 or the aggregate group: 2, 8 and 14. In
Equation 8, $vi(d)$ is the total normalized volume distribution of each aerosol component or group per volume of air.
The total volume of the aerosol group with diameters between d and d+Δd per volume of air is provided by *V(d)* in
Equation 9. $N_{N-Si}$ is the ambient value of the total near-surface aerosol number concentration for MISR
component/group *i*. The $N_{N-Si}$ value in Equation 9 will be the same as that derived directly from the MISR data in
Equations 5a and 5b only to the extent that the near-surface aerosol type represents the total-column aerosol type, an
assumption we make consistently in this analysis.
$$V_i(d) = N_{N-Si} \times \int_{d}^{d+\Delta d} v_i(d) \times d(d)$$
(9)

The integration of $v_i(d)$ for each aerosol component/group from 0 to a finite diameter is nontrivial. We solve this
integral numerically using Equation 10 to obtain the total volume contributed by each aerosol component per
volume of air. When doing this integration, we are careful to take into consideration the lower and upper limits on
the radius for each MISR aerosol component in each aerosol component/group.

$$V_i(D) = N_{N-Si} \times \sum_{d=1}^{d=D} (v_i(\frac{d}{10000}) \times 0.0001$$
(10)

The unit of volume ( $Vi$ ) here is (μm)³, as the unit of the diameter we use here is in μm. To calculate $PM_{2.5}$ we need
to multiply the total volume of each of the eight aerosol components for particles calculated using Equation 10, by
the particle density, as shown in Equation 11.
$$PM_{2.5} = density \times \sum_{i=1}^{8} V_i(D_i)$$
(11)

In this analysis, we assume the same particle density that Alphasense uses in its algorithm. We compute $PM_{2.5}$ in
units of μg/m³ from the volume obtained:
1.65 g/cm³ or $1.65 \times 10^{-6}$ μg/m³ ( $\frac{\#}{m^3} \times (10^{-18} \times \frac{m^3}{(\mu m)^3}) \times (\mu m)^3 \times 1.65 \frac{g \times \frac{10^6 \mu g}{g}}{cm^3 \times (\frac{10^{-6} m^3}{cm^3})}$ ).


Note that the Alphasense algorithm to convert particle counts to mass is proprietary, and we do not have access to its
methodology.

## 4.    Size-Dependent    Near-Surface    Particle    Concentrations,    Constrained    by Regression Against Satellite Data for Nairobi, Kenya

In this section we apply the method described in Section 3 above to the OPC and satellite data collected in Nairobi
from May 2016 through early March 2017. We present the results using the limited coincident MISR data and the
larger scaled-MODIS dataset, and then summarize the assumptions and mitigating factors in the current analysis,
which includes a discussion of possible improvements for future deployments.

### 4.1 Application of the method to the 2016-2017 Nairobi OPC deployment

Following Steps 1, 2a, and 2b of the methodology described in Section 3, Table 1 shows the near-surface AOD for
the Nairobi data obtained from the vertically scaled MISR Research Algorithm for aerosol components 1,3,6,19 and
21, as well as that for the aerosol group comprised of components 2,8 and 14, using the standard universe of 74
mixtures. Table S2 in Supplementary Information shows the lognormal size distribution (dN/d(lnD)) from the OPCs
for the coincident surface observations that correspond to the 10 successful MISR retrievals where the total $AOD_{558}$
> 0.15.

We obtain the group-specific particle-size data from MISR (Step 2c), and the associated number concentrations
($N_{N-Si}$) from Equations 5a and 5b (Step 2d). We then linked the size distribution of the MISR aerosol groups with that
of the OPCs (Step 2f).  The regression analysis was conducted using the total dN/d(lnD) derived from the MISR
measurements as the predictor of the dN/d(lnD), with the ground-based measurements as the dependent variable,
assessed at six different diameters corresponding to the mid-points of the OPC size bins Bin 2 – Bin 7 (Equation 1),
where the datasets overlap. For each of the 10 high-AOD MISR cases, we have six dN/dln(D) measurements (= 60
rows in our regression analyses).

For all regression analyses we excluded MISR component 21 as the AOD retrieved for this component is 0. In
Regression Analysis 1, we included the remaining MISR components. Not all of the coefficients in the regression
are significant, and some are negative. Each coefficient in the regression represents the total number concentration
of the respective aerosol group, which physically cannot be negative. However, it is possible for a statistical weight
to be negative, as the regression approach aims to formally match the retrieved values with available observations,
and there can be aerosol components and mixtures missing from the MISR algorithm climatology (Kahn et al.,
2010).  As such, leveraging from the better-fitting components can skew the coefficients for other particles negative.
Provided the negative weights are small compared to the dominant retrieved components, the negative values



represent noise in the results. This can apply to components 1 and 8 that are often retrieved in relatively small
quantities, as well as to component 19, a dust optical analog, that very likely does not match actual dust in the
region. Moreover, MISR component 1, with re=0.06 μm, is well below the OPC lowest size sensitivity limit.
Regression Analysis 2 was run without component 1 and 19.

The results of regression Analyses 1 and 2 are given in Table 2. Figure 1 shows the particle size distributions
(dN/dlnD) from the air quality monitors obtained for all relevant ground-based observations, superimposed on the
size distributions derived from the regression analysis results of Analysis 2. The derived size distributions from each
instrument are quite well matched in nearly all cases, despite the assumptions involved. For Analysis 2, the adjusted
R squared is 0.82.

To increase satellite sampling, we repeated the regression analysis by scaling MAIAC AODs using the monthly
effective MISR aerosol component AOD fractions (Steps 2e and 2f). We have 1712 MAIAC AOD retrievals that fall
within a radial distance of 1.6 km of a ground-station. However, there are only 10 favorable MISR particle property
retrievals, on three unique days. Using the MISR component AOD values to parse the MAIAC total-AOD, even on
a monthly basis, leaves 304 MAIAC retrievals on 20 unique days (Figure S6 in Supplementary Information). Yet
this provides about 30 times as much data as the MISR data alone. Like Analysis 1, Analysis 3 includes all MISR
aerosol components, but was run using the scaled MAIAC dataset. We also ran Analyses 4 and 5 the MAIAC data,
this time excluding MISR components 1 and 19. For Analysis 5, we also restricted the MAIAC retrievals to those
with the total AOD ≥ 0.15 (85 MAIAC AODs), to ensure that near-surface aerosols dominate in this analysis. The
adjusted R squared for Analysis 5 is 0.76. When we used MAIAC AODs at a radial distance of 1 km and 0.5 km
from each site (instead of 1.6 km), repeating Analysis 5, yielded adjusted R squared values of 0.77 in both cases.
This suggests that our results are robust to the radius considered.

The results for the five analyses are given in Table 2. All the coefficients for the remaining aerosol groups included
in Analyses 2, 4 and 5 are positive and statistically significant (p-value almost equal to, or less than 0.05). Figure 2a
shows $PM_{2.5}$ from the ground-based OPCs (scaled by a factor of 4 for the sake of comparison) and the corresponding
$PM_{2.5}$ calculated from MISR (Step 3), using the results of Analysis 2. The MISR-derived and OPC PM tend to show
similar peaks, with the exception of All Saints. Taking all points into consideration, the correlation between the two
PM datasets is 0.56. The OPC at All Saints is situated in a particularly clean area, surrounded by hotspots of
pollution due to informal settlements nearby. The average pollution in the coincident satellite grid cell is higher than
that observed by the OPC at this particular site, likely caused by the difference in spatial sampling. When we drop
measurement at All Saints from this analysis, the correlation between the derived $PM_{2.5}$ from MISR and that of the
OPC is 0.76 (Figure 2b).





Similarly, Figure S7 in Supplemental Information displays the derived $PM_{2.5}$ concentrations from MAIAC/MISR
AOD estimates using coefficients from Analysis 5 and the corresponding surface $PM_{2.5}$ from the OPCs. The
correlation between the two PM values is 0.47. When we drop All Saints, the correlation increased to 0.48.
However, the adjusted R squared is ~0.8 when working directly with size distribution information (Step 2f) rather
than the $PM_{2.5}$ values due to the additional assumptions involved (Step 3).

The satellite-derived PM values are very high relative to the OPCs in nearly all cases. An important contributing
factor is that a large fraction of aerosols in Nairobi are primary combustion aerosols with diameters < 0.54 μm that
MISR detects (Figure S4 and Table S3), but that are not included in the OPC data due to lack of sensitivity. In
addition, any secondary aerosol formation from the many sources of gaseous precursors would produce small
particles, and any underestimate in the particle density assumed in the OPC retrieval might also play a role. A
further possible contributing factor, at least at one site (Kibera Girls Soccer Academy), is the frequent dominance of
coarse mode particles, which contribute to the total AOD observed by MISR. However, MISR does not retrieve
specific size information for particles larger than about 2-3 μm (Section 2.2.1 above), so the MISR total AOD is
ascribed to smaller-sized particles, where the retrieval is sensitive; this can inflate the number concentration of these
particles. Given these issues, our method focuses on the size range over which both the OPC and MISR
measurements are sensitive (Figure 1). As most of the particles retrieved over the urban Nairobi region are
components within the typical combustion-particle size distribution (see Section S1.2.1 Supplemental Material), the
method yields a high correlation despite the limitations of the data, and actually uses the satellite data to account for
smaller particles that the OPCs miss.

**4.2 Assumptions, and mitigating factors in the current analysis, with advice for future**
**deployments**
The data collected during the 2016-2017 Nairobi experiment are not ideal for the current application. However,
there were also mitigating factors, which we summarize here, along with the lessons learned for the benefit of future
deployments.

• *MISR sampling frequency*. Generally low AOD over Nairobi, combined with the relatively narrow MISR swath
width and low latitude of the target region, left just 10 cases meeting the criteria for good aerosol-type retrievals
from MISR during the OPC surface-network deployment. As such, we were forced to assume that single or pairs of
MISR particle-type retrievals in a given month represent the aerosol properties for the entire month. However, the
observation that the MISR-retrieved particles varied little among the available observations (Figure S3) and are
typical of urban pollution from the local sources expected in Nairobi favors this approach. Selecting cases having
mid-visible AOD ≥ 0.15 also favors conditions where local sources dominate. The assumption is further supported
by GEOS-Chem model aerosol-type simulations (Section 2.3 above, and Figure S5). As AOD varies considerably
more than aerosol type at the Nairobi site, we addressed that aspect of limited MISR sampling by using MISR



monthly size-resolved information to scale the much more frequent MODIS-MAIAC AOD retrievals. In future
experiments, sites typically experiencing higher AOD, preferably also at higher latitude, as well as longer
deployments, could greatly improve the MISR sampling statistics for this application.

• *Aerosol vertical distribution*.  We also use the GEOS-Chem AOD vertical distribution to obtain the near-surface
component of the MISR total-column AOD and assume that MISR-retrieved total-column particle properties are
dominated by near-surface particles in the study region.  As expected, our analysis works best on days when the
satellite-derived AOD was ≥ 0.15, and near-surface urban aerosols dominate the column (Figure S5). The
observation that the MISR-retrieved particles are typical of urban pollution from local sources in Nairobi (Table 1
and Section 1.2.1 in Supplemental Material) also favors this assumption.  Further, dust is the most likely transported
species, and it is distinguished from pollution particles in MISR retrievals based on large size and non-spherical
shape.  AOD is derived from satellite instruments under ambient RH conditions. If the particles were hygroscopic,
however, they could adsorb water vapor and appear larger than they would be under dry conditions, which is how
$PM_{2.5}$ is usually assessed. Yet, the RH at the Nairobi site was generally low during the study period (Table S2),
pollution particles are not very hygroscopic, and the OPC measurements were also obtained at ambient RH (section
2.1 above), all mitigating the RH issue.  Unfortunately, there were no local lidar observations to validate the model
vertical aerosol distribution, and neither the CALIPSO nor the CATS space-based lidars acquired data useful for this
purpose, as discussed in section S1.3 in Supplementary Information. In future deployments, a single, well-placed
surface lidar in the region could test the assumptions about aerosol vertical distribution and determine whether any
aerosol layers aloft contribute significantly to the satellite, column-effective particle property retrievals.

• *OPC small-particle sampling*.  Pollution particles typically have diameters in the range 0.1 – 0.3 μm, and the
pollution particles MISR retrieved had effective radii 0.12 μm (effective diameter 0.24 μm). Yet, the Alphasense
OPC-N2 instruments used in the current study do not register particles < 0.38 μm in diameter, and the smallest size
bin is noisy, effectively limiting the OPC size sensitivity to particles > 0.54 μm.  As such, particle-size regressions in
this study were performed over six size bins spanning 0.54 - 3 μm, capturing the range over which both satellite and
surface instruments are sensitive.  The small-particle-observation limitations represent a significant uncertainty in
the results. However, the particle-size comparisons shown in Figure 1 demonstrate very good agreement over the
six-bin range, and further, we obtained ~0.8 $R^2$ model fits for the aerosol size distribution formally, when
considering either the MISR retrievals alone or the better-sampled MAIAC AODs parsed to the MISR component
fractions. As MISR sensitivity extends to particles ~ 0.1 μm, the satellite data help account for fine aerosols having
diameters < 0.54 μm in our analysis. For future deployments where the dominant particle type is urban pollution,
including surface instruments that have sensitivity to particles down to ~0.1 – 0.2 μm in diameter would make the
surface-station dataset substantially more robust. Further, at least one coincident, strategically located reference air
quality monitor would make it possible to quantify retrieval sensitivity with greater confidence.



## 5. Conclusions

For many locations around the world, the alternative to deploying low-cost air-quality monitors is having no ground-monitoring at all. Surface monitors are essential to help characterize the near-surface aerosol components within total-column satellite observations, but they offer only limited coverage, and the PM measurements from low-cost monitors in themselves are generally not well calibrated.

This paper develops and presents a novel method that moves away from the conventional approach of linking remotely sensed, total-column AOD from satellites with directly sampled particulate mass per volume of air from surface monitors. Instead, it combines satellite, component-specific AOD retrievals with particle counts from low-cost monitors, to constrain the size distribution of surface aerosol and $PM_{2.5}$. Retrieving some particle-size information is possible with data from the space-based MISR instrument under favorable retrieval conditions. MISR-retrieved particle effective cross-sectional area is linked with the size distribution of particulates as observed by the low-cost OPC-N2 observations. As far as we know, size-resolved particle counts have not previously been used to associate remote-sensing and direct-surface aerosol data, as most standard reference monitors provide particulate mass measurements and not particle counts partitioned by particle size.

We applied the method presented in Section 3 to data from a 2016-2017 10-month Nairobi experiment, due to the relative longevity of that data record. Limitations in the experiment design and implementation included relatively infrequent MISR sampling and low AOD, as well as the lack of a lidar or high-quality reference particle sampler in the field to validate assumptions about aerosol vertical distribution and satellite-retrieved small-particle surface concentration, respectively. However, the dominance of locally generated urban pollution particles concentrated near the surface, low relative humidity, and an effective approach for scaling more frequent MODIS data with the MISR-retrieved size distributions were mitigating factors. The method produced high correlations (~0.8) between satellite-derived and surface-station-measured $PM_{2.5}$, and most importantly, the satellite data helped significantly to account for smaller particles that tend to dominate urban aerosol pollution but are below the detection size limit of the OPCs.

Our analysis also led to specific suggestions for performing future deployments with fewer assumptions, such as including at least one carefully sited, surface-based lidar and reference air quality monitor. Applying the technique under conditions more favorable for this approach could help assess air quality in rapidly urbanizing cities in developing countries, where pollution increases are having dramatic public health consequences, and where monitoring is limited or entirely absent. We hope with the increasing focus on air quality (e.g., the expansion of the SPARTAN network, Weagle et al., 2018), broader application of low-cost monitoring can occur.




**Acknowledgements**: The authors gratefully acknowledge the United Nations Environment Program (UNEP) for
funding and piloting the low-cost air quality monitor deployment in Nairobi. In particular, many thanks go to
Jacqueline McGlade, Sami Dimassi, Valentin Foltescu and Victor Nthusi. The authors thank Colette Heald, David
Ridley for several useful discussions, Michael Garay for help with using the MISR Toolkit, John Yorks for help with
interpreting the data from CATS, Jason Tackett, Ali Omar and David Winker for help with interpreting data from
CALIPSO, and Dave Diner for inviting P. deSouza to present an early version of this work at the 2016 MISR
Science team meeting. The work of P. deSouza, F. Duarte and C.Ratti is supported by the MIT Senseable City Lab
Consortium. The work of R. Kahn is supported in part by NASA's Climate and Radiation Research and Analysis
Program under Hal Maring, as well as NASA's Atmospheric Composition Program under Richard Eckman.
**Competing Interests**
The authors declare that they have no conflict of interest





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



# Figures


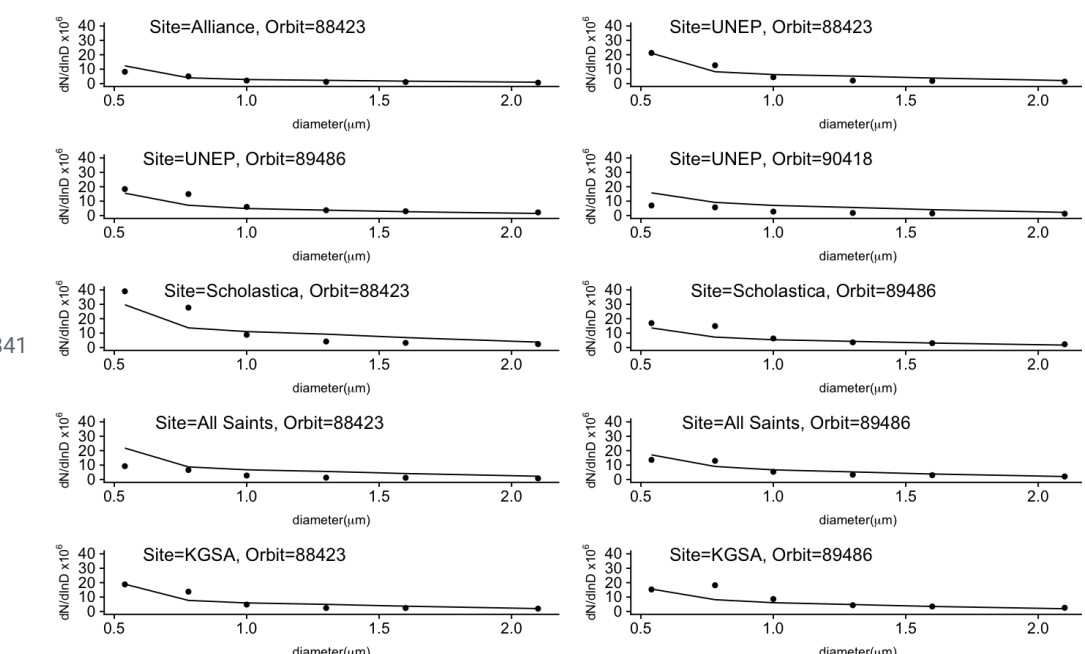



**Figure 1:** Ground-based size distributions (#/m³) obtained from the low-cost air quality monitors, represented by
points at the bin-center diameters (µm) for each of OPC bins 2-7, and the corresponding size distribution derived
from the 10 favorable MISR retrievals (represented by lines). The orbit number of the satellite observation is
provided along with which ground-based monitor location with which the satellite pixel overlapped.



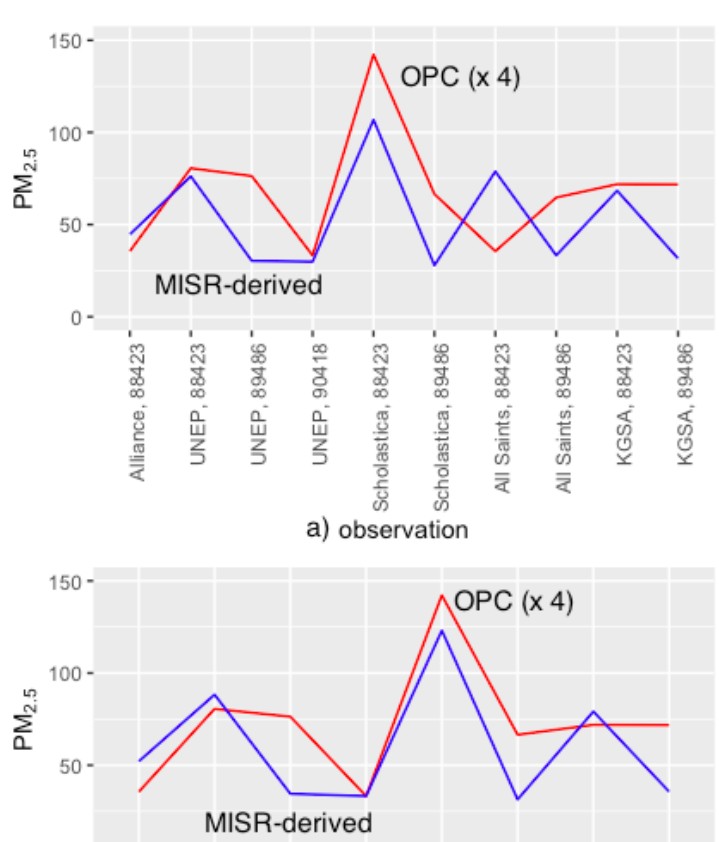



**Figure 2:** (Red) PM$_{2.5}$ (µg/m$^3$) measured from the OPC-N2 (scaled up by a factor of 4 to make comparable with the PM derived from MISR) , and (Blue) PM$_{2.5}$ calculated from coincident MISR observations for the (a) 10 cases where MISR AOD >0.15 (identified by the MISR orbit number and the coincident site name along the horizontal axis), and (b) Coincident MISR observations at all sites, but All Saints, using model coefficients from Analysis 2 in Table 2. The regression analysis yields a correlation of 0.56 for the data in panel (a), whereas the correlation is 0.76 for panel (b). A major factor contributing to the quantitative difference is probably the lack of OPC sensitivity to particles < 0.54 µm in diameter

858

859

860





# Tables

**Table 1:** Successful near-surface MISR aerosol optical depth retrievals for each MISR aerosol component (including the aggregate scaled AOD from components 2, 8 and 14), the total near-surface MISR AOD and the total MISR AOD, averaged over a radial distance of 1.6 $km^2$ from each surface monitoring site. These values were obtained for each of the 28 coincident observations from the MISR research algorithm, run with the standard universe of 74 mixtures. The AOD is set to zero for aerosol components not present among the MISR-retrieved aerosol types. The retrieved amounts of Components 19 and 21 were negligible or zero in all the retrievals. Near-surface values were obtained by scaling total-column AOD based on GEOS-Chem simulated aerosol vertical distributions. The 10 highlighted rows correspond to observations that have a MISR total AOD (sum of the AOD of the eight MISR aerosol components) > 0.15. The corresponding surface $PM_{2.5}$ from the ground-based OPC for the 10 favorable MISR retrievals is also presented. Note we have rounded the $PM_{2.5}$ values to the nearest integer to acknowledge the uncertainties in the OPC $PM_{2.5}$ measurements.

| Date | Orbit # | Location (1.6 km radial average) | MISR Near-Surface AOD by component | | | | | | Total near-surface $AOD_{558}$ | Total AOD | 30 minute-averaged OPC $PM_{2.5}$ ($\mu g/m^3$) |
| | | | 1 | 2+8+14 | 3 | 6 | 19 | 21 | | | |
|---|---|---|---|---|---|---|---|---|---|---|---|
| 8/2/16 | 88423 | UNEP | 0.00 | 0.13 | 0.00 | 0.03 | 0.00 | 0.00 | 0.156 | 0.340 | 20 |
| 8/2/16 | 88423 | Alliance | 0.00 | 0.08 | 0.00 | 0.01 | 0.00 | 0.00 | 0.090 | 0.192 | 9 |
| 8/2/16 | 88423 | Scholastica | 0.00 | 0.17 | 0.00 | 0.05 | 0.00 | 0.00 | 0.219 | 0.463 | 36 |
| 8/2/16 | 88423 | KGSA | 0.00 | 0.11 | 0.00 | 0.03 | 0.00 | 0.00 | 0.139 | 0.301 | 18 |
| 8/2/16 | 88423 | All Saints | 0.00 | 0.13 | 0.00 | 0.03 | 0.00 | 0.00 | 0.162 | 0.348 | 9 |
| 10/14/16 | 89486 | UNEP | 0.02 | 0.04 | 0.01 | 0.02 | 0.00 | 0.00 | 0.085 | 0.201 | 19 |
| 10/14/16 | 89486 | Alliance | 0.01 | 0.03 | 0.01 | 0.02 | 0.00 | 0.00 | 0.062 | 0.146 | |
| 10/14/16 | 89486 | Scholastica | 0.01 | 0.03 | 0.01 | 0.02 | 0.00 | 0.00 | 0.076 | 0.179 | 17 |
| 10/14/16 | 89486 | KGSA | 0.02 | 0.03 | 0.01 | 0.02 | 0.00 | 0.00 | 0.086 | 0.203 | 18 |



| | | | | | | | | | | | |
|---|---|---|---|---|---|---|---|---|---|---|---|
| 10/14/16 | 89486 | All Saints | 0.01 | 0.03 | 0.01 | 0.03 | 0.00 | 0.00 | 0.089 | 0.211 | 16 |
| 12/17/16 | 90418 | UNEP | 0.01 | 0.03 | 0.01 | 0.03 | 0.00 | 0.00 | 0.075 | 0.179 | 8 |
| 12/17/16 | 90418 | Alliance | 0.01 | 0.02 | 0.01 | 0.02 | 0.00 | 0.00 | 0.055 | 0.130 | |
| 12/17/16 | 90418 | Scholastica | 0.01 | 0.02 | 0.01 | 0.02 | 0.00 | 0.00 | 0.055 | 0.131 | |
| 12/17/16 | 90418 | KGSA | 0.01 | 0.01 | 0.01 | 0.01 | 0.00 | 0.00 | 0.041 | 0.102 | |
| 12/17/16 | 90418 | All Saints | 0.01 | 0.01 | 0.01 | 0.01 | 0.00 | 0.00 | 0.041 | 0.105 | |
| 1/2/17 | 90651 | KGSA | 0.01 | 0.02 | 0.01 | 0.02 | 0.00 | 0.00 | 0.048 | 0.124 | |
| 1/18/17 | 90884 | UNEP | 0.00 | 0.02 | 0.01 | 0.02 | 0.00 | 0.00 | 0.052 | 0.132 | |
| 1/18/17 | 90884 | Alliance | 0.00 | 0.02 | 0.01 | 0.01 | 0.00 | 0.00 | 0.041 | 0.106 | |
| 1/18/17 | 90884 | Scholastica | 0.00 | 0.02 | 0.01 | 0.02 | 0.00 | 0.00 | 0.047 | 0.118 | |
| 1/18/17 | 90884 | All Saints | 0.00 | 0.02 | 0.01 | 0.02 | 0.00 | 0.00 | 0.046 | 0.119 | |
| 1/25/17 | 90986 | UNEP | 0.01 | 0.02 | 0.01 | 0.02 | 0.00 | 0.00 | 0.049 | 0.123 | |
| 1/25/17 | 90986 | Scholastica | 0.01 | 0.02 | 0.01 | 0.02 | 0.00 | 0.00 | 0.046 | 0.113 | |
| 1/25/17 | 90986 | All Saints | 0.01 | 0.02 | 0.01 | 0.02 | 0.00 | 0.00 | 0.053 | 0.129 | |
| 2/3/17 | 91117 | UNEP | 0.00 | 0.00 | 0.00 | 0.00 | 0.00 | 0.00 | 0.010 | 0.028 | |
| 2/3/17 | 91117 | Alliance | 0.00 | 0.00 | 0.00 | 0.00 | 0.00 | 0.00 | 0.004 | 0.012 | |
| 2/3/17 | 91117 | Scholastica | 0.00 | 0.00 | 0.00 | 0.00 | 0.00 | 0.00 | 0.011 | 0.030 | |
| 2/3/17 | 91117 | All Saints | 0.00 | 0.01 | 0.00 | 0.01 | 0.00 | 0.00 | 0.018 | 0.049 | |
| 2/26/17 | 91452 | Alliance | 0.01 | 0.02 | 0.01 | 0.02 | 0.00 | 0.00 | 0.058 | 0.134 | |



**Table 2:** Results from multiple linear regression analyses using the size distribution of MISR aerosol components as
the independent variable, and the size distribution from the OPC as the dependent variable. In Analyses 1 and 2, the
size distribution of components for MISR observations with a total AOD> 0.15 is used. In Analyses 3, 4 and 5



MISR component AODs were obtained by scaling MAIAC AODs using the monthly effective MISR aerosol
component AOD fractions. Equations 5a and 5b are used to derive the total number concentration of each MISR
aerosol group ($N_{N-Si}$). Because the AOD retrieved for MISR aerosol component 21 is 0, we do not consider this
component in the regression analysis. Analysis 1 and 3 includes MISR aerosol component 1 and 19, while Analysis
2, 4 and 5 do not. In Analysis 5, we restricted the MAIAC retrievals considered to those where the total AOD $\geq$

883 0.15.


| | Analysis 1 (MISR only) | | Analysis 2 (MISR only) | | Analysis 3 (MAIAC) | | Analysis 4 (MAIAC) | | Analysis 5 (total MAIAC AOD $\geq$ 0.15) | |
|---|---|---|---|---|---|---|---|---|---|---|
| | Coefficients | 95% CI | Coefficients | 95% CI | Coefficients | 95% CI | Coefficients | 95% CI | Coefficients | 95% CI |
| Component1 | $-1.7 \times 10^{10}$ | $-5.1 \times 10^{10}, 1.9 \times 10^{10}$ | - | | $-3.3 \times 10^{10}$ (***) | $(-4.0, -2.6) \times 10^{10}$ | - | - | | |
| Component 2,8,14 | $4.3 \times 10^{8}$(***) | $3.2 \times 10^{8}, 5.4 \times 10^{8}$ | $4.2 \times 10^{+8}$(***) | $3.1 \times 10^{8}, 5.3 \times 10^{8}$ | $5.8 \times 10^{8}$ (***) | $(5.4, 6.2) \times 10^{8}$ | $5.3 \times 10^{8}$(***) | $(4.9, 5.7) \times 10^{8}$ | $6.0 \times 10^{8}$(***) | $(5.3, 6.6) \times 10^{8}$ |
| Component 3 | $1.4 \times 10^{9}$(*) | $0.1 \times 10^{9}, 2.6 \times 10^{9}$ | $8.2 \times 10^{+8}$(***) | $0.4 \times 10^{9}, 1.3 \times 10^{9}$ | $1.7 \times 10^{9}$(***) | $(1.5, 2.0) \times 10^{9}$ | $6.0 \times 10^{8}$(***) | $(5.0, 6.9) \times 10^{8}$ | $3.4 \times 10^{8}$(***) | $(1.6, 5.2) \times 10^{8}$ |
| Compo | 6.2 | $3.9 \times 10^{9}$, | $5.7 \times$ | $4.2 \times$ | 7.1 | $(6.4, 7.8)$ | $6.8 \times$ | $(6.4, 7.2)$ | $6.8 \times$ | $(6.1,$ |





| nent 6 | x10$^{9(***)}$ | 8.4 x 10$^9$ | 10$^{9(***)}$ | 10$^{10}$, 7.3 x 10$^{10}$ | x10$^{9(***)}$ | x 10$^9$ | 10$^{9(***)}$ | x 10$^9$ | 10$^{9(***)}$ | 7.5)x 10$^9$ |
|---|---|---|---|---|---|---|---|---|---|---|
| **Compo nent 19** | -9.0 x10$^9$ | -3.1 x 10$^{10}$, 1.2 x 10$^{10}$ | -- | -- | -1.5 x 10$^{10\,(***)}$ | (-2.2, -0.8) x 10$^{10}$ | | | - | - |
| **Adjust ed R square d** | 0.82 | | 0.82 | | 0.75 | | 0.74 | | 0.76 | |

*p-values of coefficients: 0 '\*\*\*' 0.001 '\*\*' 0.01 '\*' 0.05 '.' 0.1 ' ' 1*