# Peer review of "Combining low-cost, surface-based aerosol monitors with sizeresolved satellite data for air quality applications"

_Atmospheric Measurement Techniques, 2020_

## Referee Comment (RC1) · Anonymous Referee #1 · 19 May 2020

General Comments:

In this manuscript, the authors present a technique to combine particle counts from low-cost, ground-based sensors with the additional information provided by MISR's size resolved AOD retrieval to infer PM2.5. With some modifications, I would recommend this manuscript for publication: the technique is novel and will be of interest with scientists seeking to balance the strengths and weaknesses of low-cost sensors.

That said, there are clear limitations to the current study that may limit broader application of their approach, although many of these shortcomings are already identified by the authors. Of particular concern, but as noted by the authors, is the inability to validate their results against reference-grade observations. Without such a comparison, it is difficult to determine the relative value of this combined approach compared to the uncertainties of its underlying assumptions.

With this in mind, whether or not this work is published I would strongly encourage the authors to continue to develop this approach in a location that allows direct validation.

Specific Comments:

Supplemental L192: I have some concerns that the GEOS-Chem simulation used to scale total column AOD to near-surface AOD is based on a simulation from 2012. The amount and relative influence of transported Saharan dust and biomass burning from the Congo on the vertical distribution of aerosol have significant annual variation and may impact the author's results. A plot comparing 2012 and 2016 monthly mean MAIAC AOD over Central and Northern Africa for October and December may provide some reassurance, or alternatively motivate the need for a more recent simulation (or perhaps such a simulation could be run).

Supplemental L264: How well correlated are these results when taken against total-column AOD instead of near-surface AOD? As given, the high $r^2$ could be due to MISR AOD, even if the GEOS-Chem scaling was not working well. The change in correlation when using the total-column instead of near-surface AOD is more relevant to the quality of GEOS-Chem in this application.

Figure S5: Given the sampling shown in Table 1, it would be more useful to show the vertical structure of August and October.

What is the cause of the flat sections in the OPC PM2.5, shown in Figure S7?

At the author's discretion, it may be appropriate to mention the application of such a technique to the upcoming MAIA mission. I expect MAIA's multi-angular viewing will allow similar size-resolved information as MISR provides. If appropriate, this connection would help broaden the applicability of the author's work.

---

## Referee Comment (RC2) · Anonymous Referee #2 · 11 Jun 2020

Review of 'Combining low-cost, surface-based aerosol monitors with size-resolved satellite data for air quality applications' by Priyanka de Souza

This work deals with the combination of low-cost sensors combined with satellite data to obtain PM2.5 near surface. The novelty of this technique has no doubt and the implications in the aerosol science community are huge. Authors include the shortcomings and other issues related to the technique. Methodology is well described too.

However, I have concerns before recommending the final publications. As the other referee suggests and even the authors admit, the technique needs evaluation versus other instrumentation that provide accurate PM2.5 measurements. Authors must pro-

vide at least a intercomparisons of low-cost sensors with reference instrumentation and provide a plan for future evaluations of the methodology in places with more advanced instrumentation.

I have also other concerns:

With the hypothesis related to MISR retrievals and aerosol vertical distribution, why not doing intercomparisons directly with MERRA-2 data?

What are the peculiarities of Alphasense OPC versus other low-cost sensors?

It is difficult to follow the methodology section. At least a Flow chart is needed. Also, I get confused in the intercomparisons because you make mention to number concentration in MISR and mass concentration with the sensors. That must be clarified.

The results section is not clear. Much information from the supplement must be included in the paper as supplement seems an independent paper.

Minor concerns:

Line 46: Latest development in technology has reduced the cost of accurate instrumentation. Please, be careful

Line 73: Be aware that new satellites are improving the spatial resolution

Line 110: Please, add references.

Line 212: What uncertainties are you referring to?

Line 220: MODIS also assumes certain aerosol types and can provide an estimate of particle size distribution. Please check

Lines 244-245: Why do you need gases from GEOS-Chem? Please avoid unnecessary information because paper is already too long.

Lines 353-354: AOD is by definition over the vertical, so your definition is not correct. Are you referring to aerosol optical thickness? Please correct. Lines 442-448: Here is

what I do not understand about particle density. Why do you need that?

Results: I do not understand what do you mean about Analysis 1, 2, 3, 4 y 5

Tables 1 and Tables 2 need further explanations.

―――――――――――――――――

---

## Referee Comment (RC3) · Anonymous Referee #3 · 17 Jul 2020

General comments

It is not clear the general scope of the manuscript. it seems that an older draft has been readapted for new purposes. From the title I would expect that the performances of new low-cost sensors in monitoring aerosols are assessed and supported by satellite measurements. Rather, the satellite observations are needed to improve low-cost sensor performances and extend its measurement range. This is pretty unusual. Usually it is the other way round. Satellite observations are at much coarser resolution.

The authors are however aware that considering the monthly effective fraction doesn't make so much sense. In-situ measurements can catch a variability that is order of
magnitude higher. Moreover, OPC can't detect aerosols with a diameter smaller than 0.38 micrometers. Exhaust and combustion aerosol size is much lower that that value.

In the paper some statements are not state-of-the-art and should be corrected. Technology made progress in the last years and cheaper reliable instruments are available nowadays. This reminds the observations stated in the first comment.

The presented methodology might be interesting, but the same experiment should be repeated where lidar and sun-photometer measurements are available. Why developing a technique in a place where it cannot be properly validated ? There is an agreement between MISR-MAIAC and in-situ sensor, but this tells us nothing if the retrievals are accurate I would perform the same analysis at NASA Goddard to prove true those claims.

Specific comments can be found in the attached file.

Please also note the supplement to this comment:
https://amt.copernicus.org/preprints/amt-2020-136/amt-2020-136-RC3-supplement.pdf
* * *
[Figure]

**Supplement:**

[revised manuscript text omitted]

towards a mechanistic model of global soil nitric oxide emissions: implementation and space based-constraints.
Atmospheric Chemistry & Physics 12, 7779–7795. https://doi.org/10.5194/acp-12-7779-2012
Kahn, R., Banerjee, P., McDonald, D., 2001. Sensitivity of multiangle imaging to natural mixtures of aerosols over
ocean. Journal of Geophysical Research: Atmospheres 106, 18219–18238. https://doi.org/10.1029/2000JD900497
Kahn, R.A., Gaitley, B.J., 2015. An analysis of global aerosol type as retrieved by MISR. Journal of Geophysical
Research: Atmospheres 120, 4248–4281. https://doi.org/10.1002/2015JD023322
Kahn, R.A., Gaitley, B.J., Garay, M.J., Diner, D.J., Eck, T.F., Smirnov, A., Holben, B.N., 2010. Multiangle Imaging
SpectroRadiometer global aerosol product assessment by comparison with the Aerosol Robotic Network. Journal of
Geophysical Research: Atmospheres 115. https://doi.org/10.1029/2010JD014601
Kumar, P., Morawska, L., Martani, C., Biskos, G., Neophytou, M., Di Sabatino, S., Bell, M., Norford, L., Britter, R.,
2015. The rise of low-cost sensing for managing air pollution in cities. Environment International 75, 199–205.
https://doi.org/10.1016/j.envint.2014.11.019
Lee, H.J., Liu, Y., Coull, B.A., Schwartz, J., Koutrakis, P., 2011. A novel calibration approach of MODIS AOD data
to predict PM 2.5 concentrations. https://doi.org/10.5194/acp-11-7991-2011
Levy, R.C., Mattoo, S., Munchak, L.A., Remer, L.A., Sayer, A.M., Patadia, F., Hsu, N.C., 2013. The Collection 6
MODIS aerosol products over land and ocean. Atmospheric Measurement Techniques 6, 2989–3034.
https://doi.org/10.5194/amt-6-2989-2013
Lewis, A., Edwards, P., 2016. Validate personal air-pollution sensors. Nature 535, 29–31.
https://doi.org/10.1038/535029a
Limbacher, J.A., Kahn, R.A., 2017. Updated MISR dark water research aerosol retrieval algorithm – Part 1: Coupled
1.1 km ocean surface chlorophyll *a* retrievals with empirical calibration corrections. Atmospheric Measurement
Techniques 10, 1539–1555. https://doi.org/10.5194/amt-10-1539-2017
Limbacher, J.A., Kahn, R.A., 2015. MISR empirical stray light corrections in high-contrast scenes. Atmospheric
Measurement Techniques 8, 2927–2943. https://doi.org/10.5194/amt-8-2927-2015
Limbacher, J.A., Kahn, R.A., 2014. MISR Research Aerosol Algorithm: refinements for dark water retrievals.
Atmospheric Measurement Techniques Discussions 7, 7837–7882. https://doi.org/10.5194/amtd-7-7837-2014
Limbacher, J., Kahn, R.A., 2019. Updated MISR Over-Water Research Aerosol Retrieval Algorithm - Part 2: A
Multi-Angle Aerosol Retrieval Algorithm for Shallow, Turbid, Oligotrophic, and Eutrophic Waters. Atmospheric
Measurement Techniques 675–689. https://doi.org/10.5194/amt-12-675-2019,
http://dx.doi.org/10.5194/amt-12-675-2019

[revised manuscript text omitted]

---

## Author Comment (AC1) · 30 Jul 2020

We are grateful for the constructive reviews we received for our paper. We have modified the manuscript to address the reviewers' comments, and herein resubmit the updated paper and detailed responses to the reviewers.

Reviewer 1

General Comments: In this manuscript, the authors present a technique to combine particle counts from low-cost, ground-based sensors with the additional information provided by MISR's size resolved AOD retrieval to infer PM2.5. With some modifica-

tions, I would recommend this manuscript for publication: the technique is novel and will be of interest with scientists seeking to balance the strengths and weaknesses of low-cost sensors.

Thank you. We are grateful to the reviewer for recognizing the novelty of this technique.

That said, there are clear limitations to the current study that may limit broader application of their approach, although many of these shortcomings are already identified by the authors. Of particular concern, but as noted by the authors, is the inability to validate their results against reference-grade observations. Without such a comparison, it is difficult to determine the relative value of this combined approach compared to the uncertainties of its underlying assumptions. With this in mind, whether or not this work is published I would strongly encourage the authors to continue to develop this approach in a location that allows direct validation.

We agree with the reviewer. In the current paper, we describe a novel methodology and demonstrate it using data from Nairobi. We recognize that many conditions of that experiment were not ideal. Our demonstration of this technique using data from Nairobi helps us highlight these limitations, which are enumerated in the text and supplement, as the reviewer acknowledges. We aim to use the publication of the current paper, presenting the technique, along with what we have learned from this initial pilot, to support a proposal for a future deployment that will allow us to validate this methodology under more ideal conditions. The published paper giving the technique, along with the limitations of the Nairobi experiment, will be essential support for any proposal we might write requesting to perform an improved experiment.

Specific Comments:

Supplemental L192: I have some concerns that the GEOS-Chem simulation used to scale total column AOD to near-surface AOD is based on a simulation from 2012. The amount and relative influence of transported Saharan dust and biomass burning from the Congo on the vertical distribution of aerosol have significant annual variation

and may impact the author's results. A plot comparing 2012 and 2016 monthly mean MAIAC AOD over Central and Northern Africa for October and December may provide some reassurance, or alternatively motivate the need for a more recent simulation (or perhaps such a simulation could be run).

This is a fair point. Unfortunately, more recent GEOS-Chem simulations are not available to us for Nairobi, nor the ability to re-run the model. Thus, the purpose of this paper is limited to methods development. In the future, we hope to validate this method in a more ideal location: one for which we will have CALIPSO or other space- and/or ground-based lidar data, a collocated reference monitor, along with a contemporaneous run of the GEOS-Chem or other model, to assess aerosol vertical distribution with greater confidence. We have discussed this in Section 4.2 of the paper

Supplemental L264: How well correlated are these results when taken against total column AOD instead of near-surface AOD? As given, the high rËĘ2 could be due to MISR AOD, even if the GEOS-Chem scaling was not working well. The change in correlation when using the total-column instead of near-surface AOD is more relevant to the quality of GEOS-Chem in this application.

When repeating this analysis with the total column AOD for the 10 measurements, we obtain an adjusted R squared of 0.89. This is comparable with the adjusted R squared obtained when using the near-surface AOD (0.88). This supports the assumption that the aerosol is concentrated near-surface. We have included a short sentence in the text in Section S2 in the SI about this.

Figure S5: Given the sampling shown in Table 1, it would be more useful to show the vertical structure of August and October.

Thank you. We have updated this Figure.

What is the cause of the flat sections in the OPC $PM_{2.5}$, shown in Figure S7?

The flat estimates are because a single OPC $PM_{2.5}$ value was used to constrain MA-
IAC AODs of the grid cells within a 1.6 km radius from each surface monitoring site. Thus, the same $PM_{2.5}$ value from an OPC is linked with multiple MAIAC-derived $PM_{2.5}$ concentrations.

We have added this information to the caption of Figure S7. Thank you. The updated caption is reproduced here:

"Figure S7: (Blue) $PM_{2.5}$ values (in $\mu$g/$m^3$) from the MAIAC Analysis 5 in Table 2 (Remember only 85 satellite observations with the total MAIAC AOD $\geq$ 0.15 are considered in this analysis). The corresponding daily-averaged $PM_{2.5}$ from the ground-based OPC in units of $\mu$g/$m^3$ are shown in red. The correlation between the two estimates of $PM_{2.5}$ is 0.47. Note that the flat estimates are because a single OPC $PM_{2.5}$ value was used to calibrate MAIAC AODs of the grid cells within a 1.6 km radius from each surface monitoring site. Thus, the same $PM_{2.5}$ value from an OPC is linked with multiple MAIAC-derived $PM_{2.5}$ concentrations."

At the author's discretion, it may be appropriate to mention the application of such a technique to the upcoming MAIA mission. I expect MAIA's multi-angular viewing will allow similar size-resolved information as MISR provides. If appropriate, this connection would help broaden the applicability of the author's work.

We are aware of MAIA, and now mention this possibility in the text. Note that to apply our method, we would also need to deploy OPCs at one or more locations that MAIA is sampling. Specifically, we include the following text in the Conclusion:

"We hope with the increasing focus on air quality (e.g., the expansion of the SPARTAN network, Weagle et al., 2018), broader application of low-cost monitoring can occur. Further, the planned MAIA instrument (expected launch year: 2022), like MISR, will be able to provide size-resolved information about aerosols from space for a subset of cities at higher temporal resolution (Diner et al., 2018). As such, it should better capture the variability in aerosol type, and the data can be incorporated into our methodology."

---

## Author Response (AR1)

**Response to Reviews**

We are grateful for the constructive reviews we received for our paper. We have modified the manuscript to address the reviewers' comments, and herein resubmit the updated paper and detailed responses to the reviewers. We have highlighted the changes we made in our paper in response to comments.

**Reviewer 1**

General Comments: In this manuscript, the authors present a technique to combine particle counts from low-cost, ground-based sensors with the additional information provided by MISR's size resolved AOD retrieval to infer PM2.5. With some modifications, I would recommend this manuscript for publication: the technique is novel and will be of interest with scientists seeking to balance the strengths and weaknesses of low-cost sensors.

**Thank you. We are grateful to the reviewer for recognizing the novelty of this technique.**

That said, there are clear limitations to the current study that may limit broader application of their approach, although many of these shortcomings are already identified by the authors. Of particular concern, but as noted by the authors, is the inability to validate their results against reference-grade observations. Without such a comparison, it is difficult to determine the relative value of this combined approach compared to the uncertainties of its underlying assumptions. With this in mind, whether or not this work is published I would strongly encourage the authors to continue to develop this approach in a location that allows direct validation.

We agree with the reviewer. In the current paper, we describe a novel methodology and demonstrate it using data from Nairobi. We recognize that many conditions of that experiment were not ideal. Our demonstration of this technique using data from Nairobi helps us highlight these limitations, which are enumerated in the text and supplement, as the reviewer acknowledges. We aim to use the publication of the current paper, presenting the technique, along with what we have learned from this initial pilot, to support a proposal for a future deployment that will allow us to validate this methodology under more ideal conditions. The published paper giving the technique, along with the limitations of the Nairobi experiment, will be essential support for any proposal we might write requesting to perform an improved experiment.

**Specific Comments:**

Supplemental L192: I have some concerns that the GEOS-Chem simulation used to scale total column AOD to near-surface AOD is based on a simulation from 2012. The amount and relative influence of transported Saharan dust and biomass burning from the Congo on the vertical distribution of aerosol have significant annual variation and may impact the author's results. A plot comparing 2012 and 2016 monthly mean MAIAC AOD over Central and Northern Africa for October and December may provide some reassurance, or alternatively motivate the need for a more recent simulation (or perhaps such a simulation could be run).

This is a fair point. Unfortunately, more recent GEOS-Chem simulations are not available to us for Nairobi, nor the ability to re-run the model. Thus, the purpose of this paper is limited to

methods development. In the future, we hope to validate this method in a more ideal location: one for which we will have CALIPSO or other space- and/or ground-based lidar data, a collocated reference monitor, along with a contemporaneous run of the GEOS-Chem or other model, to assess aerosol vertical distribution with greater confidence.

Supplemental L264: How well correlated are these results when taken against total column AOD instead of near-surface AOD? As given, the high r2 could be due to MISR AOD, even if the GEOS-Chem scaling was not working well. The change in correlation when using the total-column instead of near-surface AOD is more relevant to the quality of GEOS-Chem in this application.

When repeating this analysis with the total column AOD for the 10 measurements, we obtain an adjusted R squared of 0.89. This is comparable with the adjusted R squared obtained when using the near-surface AOD (0.88). This supports the assumption that the aerosol is concentrated near-surface. We have included a short sentence in the text in Section S2 in the SI.

Figure S5: Given the sampling shown in Table 1, it would be more useful to show the vertical structure of August and October.

Thank you.

What is the cause of the flat sections in the OPC PM2.5, shown in Figure S7?

The flat estimates are because a single OPC  $PM_{2.5}$  value was used to constrain MAIAC AODs of the grid cells within a 1.6 km radius from each surface monitoring site. Thus, the same  $PM_{2.5}$  value from an OPC is linked with multiple MAIAC-derived  $PM_{2.5}$  concentrations.

We have added this information to the caption of Figure S7. Thank you.

At the author's discretion, it may be appropriate to mention the application of such a technique to the upcoming MAIA mission. I expect MAIA's multi-angular viewing will allow similar size-resolved information as MISR provides. If appropriate, this connection would help broaden the applicability of the author's work.

We are aware of MAIA, and now mention this possibility in the text. Note that to apply our method, we would also need to deploy OPCs at one or more locations that MAIA is sampling. Specifically, we include the following text in the Conclusion:

"We hope with the increasing focus on air quality (e.g., the expansion of the SPARTAN network, Weagle et al., 2018), broader application of low-cost monitoring can occur. Further, the planned MAIA instrument (expected launch year: 2022), like MISR, will be able to provide size-resolved information about aerosols from space for a subset of cities at higher temporal resolution (Diner et al., 2018). As such, it should better capture the variability in aerosol type, and the data can be incorporated into our methodology. "

**Reviewer 2**

This work deals with the combination of low-cost sensors combined with satellite data to obtain PM2.5 near surface. The novelty of this technique has no doubt and the implications in the aerosol science community are huge. Authors include the shortcomings and other issues related to the technique. Methodology is well described too.

**Thank you**

However, I have concerns before recommending the final publications. As the other referee suggests and even the authors admit, the technique needs evaluation versus other instrumentation that provide accurate PM2.5 measurements.

Authors must provide at least an intercomparisons of low-cost sensors with reference instrumentation and provide a plan for future evaluations of the methodology in places with more advanced instrumentation.

Thank you for this comment. We cite in the paper previous inter-comparisons of the OPC-N2 we used with reference equipment elsewhere. Specifically, we note:

"Sousan et al. (2016) discuss the accuracy of these [OPC] count measurements in detail, and note that they agree well with reference instrument measurements for coarser particles (> 0.78  $\mu$ m in diameter), but underestimate the particle counts for finer particles." As Nairobi did not have a reference monitor at the time of the OPC deployment, it is impossible for us to do such an inter-comparison within the current study.

Specifically, we say:

"Co-locating the OPC with a reference monitor to obtain high-quality PM data would be required to calibrate the raw OPC measurements and distinguish the signal from noise directly. However, this would be costly and possibly time-consuming (Castell et al., 2017; Rai et al., 2017). Due to limited resources, and lack of access to a reference monitor, we were unable to co-locate our low-cost sensors with a reference monitor in Nairobi at the time of the experiment. As such, we rely primarily upon the more robust raw particle counts per size bin reported by the monitors, rather than the reported  $PM_{2.5}$ ."

We agree that this is a limitation, and we make this clear in the paper. As also stated in the paper, we need to repeat this experiment under more ideal conditions; we enumerate the key improvements required in the Discussion section of our paper.

I have also other concerns: With the hypothesis related to MISR retrievals and aerosol vertical distribution, why not doing intercomparisons directly with MERRA-2 data? What are the peculiarities of Alphasense OPC versus other low-cost sensors?

The MISR data have been extensively validated, and the Research Algorithm results for particle properties, in particular, are among the best available. Comparing the MISR results directly with MERRA-2 in a meaningful way for the current application would require assuming the mass-extinction efficiency (MEE) of the particles, which is uncertain to factors of three or more.

It is difficult to follow the methodology section. At least a Flow chart is needed.

Thank you. We have included the flow-chart below at the beginning of the Methods section.

**Step 1**: Estimate the ground-based size distribution of aerosols at each site from the Alphasense OPC-N2 monitors

**Step 2**: Estimate stable and consistent aerosol size-resolved information from satellite data

Step 2a: Estimate the near-surface fraction of satellite AODStep 2b: Associate the near-surface AOD with particular aerosol species in the model

**Step 2c**: Derive the satellite-component size distribution contributions to the same size ranges as the OPC-N2 bins

**Step 2d**: Formulate the satellite constraint on size-specific surface concentration so it can be regressed against the OPC data

**Step 2e**: Increase the number of satellite data points by scaling MODIS AOD with MISR sizes

**Step 2f:** Regress the satellite near-surface, size-constrained particle concentration constraints against the OPC data to obtain a more complete near-surface aerosol size-concentration distribution

**Step 3**: Calculate  $PM_{2.5}$  from the number concentration of different MISR aerosol groups

Also, I get confused in the intercomparisons because you make mention to number concentration in MISR and mass concentration with the sensors. That must be clarified. The results section is not clear. Much information from the supplement must be included in the paper as supplement seems an independent paper.

The optical measurements from MISR allow us to derive column AODs and particle size distributions. Given the vertical aerosol distribution constraints from reanalysis, we can deduce particle number concentrations.

The PM2.5 criteria pollutant is technically the mass of near-surface particles with diameter <2.5  $\mu$ m. The OPC-N2 also makes optical measurements. The proprietary software from the company assumes particle density, in order to convert the number concentration of aerosols each size bin to a mass concentration. Thus, we assume particle density as well, to compare the MISR result with the OPCs.

Regarding the Supplement, based on other reviewer comments, we make a clear distinction between the presentation of the Method, appropriate for the AMT journal and given in the main paper, and the Nairobi experiment, which represents a loose demonstration of the method, but not a validation. Given the limitations of the Nairobi data for validating the method formally, we include that analysis in the Supplement. This keeps the work available to interested readers, but avoids leaving any impression that the Nairobi experiment in itself should be considered an adequate test of the method. Minor concerns: Line 46: Latest development in technology has reduced the cost of accurate instrumentation. Please, be careful

Thank you. We have changed this sentence to read:

"This is because air quality monitoring equipment tends to be costly to purchase **(capital costs are in the range of several thousand of US dollars**) and maintenance, and data processing and analysis requires additional expertise and resources (deSouza, 2017; Kumar et al., 2015; Mead et al., 2013)."

Line 73: Be aware that new satellites are improving the spatial resolution

Thank you for this note

Line 110: Please, add references.

**Thank you. We have included a reference**

Line 212: What uncertainties are you referring to?

The uncertainties in MISR-retrieved aerosol particle properties are assessed based on the range of particle size, SSA, and fraction non-spherical values among the aerosol mixtures from the algorithm climatology that pass the acceptance criteria (Kahn and Gaitley, 2015). There is additional uncertainty due to any limitations in the range of particle types in the assumed climatology, though the MISR Research Algorithm has an especially rich climatology (Limbacher and Kahn, 2014).

We have added this information to line 212.

Please check Lines 244-245: Why do you need gases from GEOS-Chem? Please avoid unnecessary information because paper is already too long.

**Thank you we have deleted this information**

Lines 353-354: AOD is by definition over the vertical, so your definition is not correct. Are you referring to aerosol optical thickness? Please correct.

**We have modified the text in the paper to read:**

By definition,  $AOD_{558}$  is proportional to [the number concentration of aerosols] x [the extinction area of each particle at 558 nm wavelength] x [the path over which AOD is assessed (which for MISR is the entire column. Here, we scale the AOD to provide the near-surface component residing in the lowest layer of the GEOS-Chem model, which is 130 meters vertically)].

Lines 442-448: Here is what I do not understand about particle density. Why do you need that?

Particle density is required to relate the particle volume, which is constrained optically from MISR, with particle mass, which is measured by the OPCs. (See our previous answer to this question)

Results: I do not understand what do you mean about Analysis 1, 2, 3, 4 y 5 Tables 1 and Tables 2 need further explanations.

Table 1 provides the successful co-incident MISR retrievals corresponding to each surface site for the duration of the experiment. MISR retrievals where the total AOD >=0.15 (indicating a favorable retrieval where MISR is able to distinguish between 3-5 bins in column-effective particle size) are highlighted.

**In the text we add:**

"Table 1 shows the near-surface AOD for the Nairobi data obtained from the vertically scaled MISR Research Algorithm results for aerosol components 1,3,6,19 and 21, as well as that for the aerosol group comprised of components 2,8 and 14, using the standard universe of 74 mixtures. Near-surface values were obtained by scaling total-column AOD based on GEOS-Chem simulated aerosol vertical distributions. The 10 highlighted rows correspond to observations that have a MISR total AOD (sum of the AOD of the eight MISR aerosol components) > 0.15. The corresponding surface PM2.5 from the ground-based OPC for the 10 favorable MISR retrievals is also presented."

**And elsewhere:**

We have performed multiple analyses making different assumptions, to explore the range of impacts these choices have on the results. The different analyses are summarized here:

- 1) Analysis 1: We only consider observations from MISR, for all MISR aerosol components except for component 21
- 2) Analysis 2: We only consider observations from MISR, for all components except for components 1 and 21
- 3) Analysis 3: We consider the scaled MAIAC AODs for all MISR components except 21
- 4) Analysis 4: We considered scaled MAIAC AODs for all components except 1 and 21
- 5) Analysis 5: We considered scaled MAIAC AODs where the total MAIAC AOD >= 0.15, for all components except 1 and 21"

In order to make this clearer, we have reorganized the section 4.1 to read as follows:

**"4.1.1 Only MISR retrievals considered (Analyses 1 and 2)**

For all regression analyses we excluded MISR component 21 as the AOD retrieved for this component is 0.

In Regression Analysis 1, we included the remaining MISR components. Not all of the coefficients in the regression are significant, and some are negative. Each coefficient in the regression represents the total number concentration of the respective aerosol group, which physically cannot be negative. However, it is possible for a statistical weight to be negative, as the regression approach aims to formally match the retrieved values with available observations, and there can be aerosol components and mixtures missing from the MISR algorithm climatology (Kahn et al., 2010). As such, leveraging from the better-fitting

components can skew the coefficients for other particles negative. Provided the negative weights are small compared to the dominant retrieved components, the negative values represent noise in the results. This can apply to components 1 and 8 that are often retrieved in relatively small quantities, as well as to component 19, a dust optical analog, that very likely does not match actual dust in the region. Moreover, MISR component 1, with re=0.06 µm, is well below the OPC lowest size sensitivity limit.

Regression Analysis 2 was thus run without component 1 and 19.

The results of regression Analyses 1 and 2 are given in Table 2. Figure 2 shows the particle size distributions (dN/dlnD) from the air quality monitors obtained for all relevant ground-based observations, superimposed on the size distributions derived from the regression analysis results of Analysis 2. The derived size distributions from each instrument are quite well matched in nearly all cases, despite the assumptions involved. The Nairobi aerosol has a size distribution that is sampled by MISR. The large-end tail is sampled by the OPCs, and our method uses the region of size-overlap to perform the particle-size scaling, The results in Figure 2 indicate that the two instruments are in fact sampling parts of the same particle size distribution. For Analysis 2, the adjusted R squared is 0.82.

**4.1.2 Using scaled-MAIAC retrievals (Analyses 3, 4, and 5)**

To increase satellite sampling, we repeated the regression analysis by scaling MAIAC AODs using the monthly effective MISR aerosol component AOD fractions (Steps 2e and 2f). We have 1712 MAIAC AOD retrievals that fall within a radial distance of 1.6 km of a ground-station. However, there are only 10 favorable MISR particle property retrievals, on three unique days. Using the MISR component AOD values to parse the MAIAC total-AOD, even on a monthly basis, leaves 304 MAIAC retrievals on 20 unique days (Figure S6 in Supplementary Information). Yet this provides about 30 times as much data as the MISR data alone.

Like Analysis 1, Analysis 3 includes all MISR aerosol components, but was run using the scaled MAIAC dataset. We also ran Analyses 4 and 5 with the MAIAC data, this time excluding MISR components 1 and 19. For Analysis 5, we further restricted the MAIAC retrievals to those with the total AOD  $\geq$  0.15 (85 MAIAC AODs), to ensure that near-surface aerosols dominate in this analysis. The adjusted R squared for Analysis 5 is 0.76. When we used MAIAC AODs at a radial distance of 1 km and 0.5 km from each site (instead of 1.6 km), repeating Analysis 5, yielded adjusted R squared values of 0.77 in both cases. This suggests that our results are robust to the radius considered.

The results for the five analyses are given in Table 2. All the coefficients for the remaining aerosol groups included in Analyses 2, 4 and 5 are positive and statistically significant (p-value almost equal to, or less than 0.05)."

**Anonymous Referee #3**

General comments It is not clear the general scope of the manuscript. it seems that an older draft has been readapted for new purposes. From the title I would expect that the performances of new low-cost sensors in monitoring aerosols are assessed and supported by satellite measurements. Rather, the satellite observations are needed to improve low-cost sensor performances and extend its measurement range. This is pretty unusual. Usually it is the other way round. Satellite observations are at much coarser resolution.

We appreciate that the reviewer grasps the novelty of our method. The purpose of the manuscript is to develop a novel technology to use the size distribution from the low-cost OPC-N2s to constrain the measurements from MISR (which provides stable and consistent size resolved information over time) for the aerosol size range which is visible to both instruments. As the OPC-N2s cannot reliably detect particles of sizes smaller than 0.38  $\mu$ m, we use the constrained MISR size-resolve information to improve the OPC estimates.

The low-cost OPCs have well known limitations, and we present an approach that uses MISRretrieved particle properties to address some of those limitations. For this application, the MISR Research Algorithm aerosol data offer sufficiently high spatial resolution (1.1 km pixels). They can provide meaningful information over any large urban area, as has been demonstrated previously in papers; for example, Patadia et al. (2012) applied the MISR RA results over Mexico City.

The authors are however aware that considering the monthly effective fraction doesn't make so much sense. In-situ measurements can catch a variability that is order of magnitude higher.

Right. However (1) the only *in situ* data available for the Nairobi experiment are from the OPCs, which we use to the extent possible, and (2) the Nairobi experiment itself has many issues, which we enumerate. In particular, the low-latitude location of the city, and the generally low AOD during the study period, limit the number of available, good-quality MISR observations.

This is the best we can do with the current data, and is one reason why the experiment is relegated to Supplemental Material: to make a clear distinction between the presentation of the method in the main text, appropriate for the AMT journal, and the Nairobi experiment, which represents a loose demonstration of the method and not a validation. This is also why we enumerate the limitations of that experiment, and why we call for a future experiment that addresses these limitations.

With the planned deployment of MAIA, which can provide retrievals at a much higher temporal resolution, this limitation can be addressed in future experiments. We note this in the Conclusions of the main document. Specifically, we say:

"We hope with the increasing focus on air quality (e.g., the expansion of the SPARTAN network, Weagle et al., 2018), broader application of low-cost monitoring can occur. **Further, the planned MAIA instrument (expected launch year: 2022), like MISR, will be able to provide sizeresolved information about aerosols from space for a subset of cities at higher temporal resolution (Diner et al., 2018). As such, it should better capture the variability in aerosol type, and the data can be incorporated into our methodology.** "

Moreover, OPC can't detect aerosols with a diameter smaller than 0.38 micrometers. Exhaust and combustion aerosol size is much lower that that value.

We are of course aware of that too, and we say so in Section 2.1 of the paper. The key here is that the aerosol in Nairobi has a size distribution that is sampled by MISR, and the large-end tail of which is sampled by the OPCs. We use the region of size-overlap to perform the particle-size scaling, and the result is given in Figure 1 of the paper. Despite the limitations, the results are

quite reasonable, which indicates that the two instruments are in fact sampling parts of the same particle size distribution.

In the paper some statements are not state-of-the-art and should be corrected. Technology made progress in the last years and cheaper reliable instruments are available nowadays. This reminds the observations stated in the first comment.

Thank you. We have altered our estimates of how much a reference monitor costs: "This is because air quality monitoring equipment tends to be costly to purchase **(capital costs are in the range of several thousand of US dollars**) and maintenance, and data processing and analysis requires additional expertise and resources (Kumar et al., 2015; Mead et al., 2013)."

The presented methodology might be interesting, but the same experiment should be repeated where lidar and sun-photometer measurements are available. Why developing a technique in a place where it cannot be properly validated ? There is an agreement between MISR-MAIAC and in-situ sensor, but this tells us nothing if the retrievals are accurate I would perform the same analysis at NASA Goddard to prove true those claims.

The MISR AOD and Research Algorithm particle properties have been validated extensively in previous papers cited in the current paper in section 2.2.1. There would be no point in repeating that work here.

When the Nairobi experiment was designed and performed, its effectiveness and its limitations were not known. The current paper represents an effort to develop a technique that makes use of low-cost sensors for meaningful air quality monitoring. In the process of developing this technique and applying it to the Nairobi data, we identified the limitations of the Nairobi experiment very specifically in Section 4.3. This puts us in a position to at least propose an experiment that addresses the limitations, can be used to formally validate the technique, and we hope, be applied more widely, especially where air quality monitoring is very limited or entirely absent due to the high cost of reference monitors.

Specific comments identified by the reviewer in the Supplemental Information:

Line 33: In the Abstract we say:

"We thus identify factors that will reduce the uncertainty in this approach for future experiments." The reviewer notes: "There is not ground truth observations to assess the uncertainty"

We agree with you, that in the future we need to repeat our experiment with better ground-truth observations. We spell out what these improvements should be in Section 4.3, where we stress the importance of repeating the experiment with co-location with a reference monitor, and with a surface lidar instrument.

Line 45-48: The reviewer notes that we overestimate the price of a reference monitor Thank you. We have modified the text, see our previous answer.

Line 55: One of the major drawbacks of using the lower-cost sensors is that no standards or certification criteria exist for these instruments yet, and consequently, the quality of the data they produce is of s**pecial concern**

The reviewer notes re the term special concern: still, this statement is strong. Low-cost instruments also assure quality measurements. Carotenuto, F.; Brilli, L.; Gioli, B.; Gualtieri, G.; Vagnoli, C.; Mazzola, M.; Viola, A.P.; Vitale, V.; Severi, M.; Traversi, R.; Zaldei, A. Long-Term Performance Assessment of Low-Cost Atmospheric Sensors in the Arctic Environment. *Sensors* **2020**, *20*, 1919.

Cavaliere, A.; Carotenuto, F.; Di Gennaro, F.; Gioli, B.; Gualtieri, G.; Martelli, F.; Matese, A.; Toscano, P.; Vagnoli, C.; Zaldei, A. Development of Low-Cost Air Quality Stations for Next Generation Monitoring Networks: Calibration and Validation of PM2.5 and PM10 Sensors. *Sensors* **2018**, *18*, 2843.

Thank you, we have modified the sentence to read:

"One of the major drawbacks of using the lower-cost sensors is that no standards or certification criteria exist for these instruments yet, and consequently, the quality of the data they produce is of concern, **although the performance of such instruments has been improving** (Carotenuto et al., 2020; Cavaliere et al., 2018; EPA, 2016; Lewis and Edwards, 2016)"

Line 73: The reviewer says "I doubt that low cost instruments can be effective in aerosol speciation"

Thank you. In this line we were referring to challenges of using satellite-derived AOD information and were speaking of MISR's ability to discriminate between the different aerosol components. We do not believe that low-cost sensors can discriminate between aerosol types. We only use the ability of the OPC-N2s to report particle number concentrations in different size bins in detailing this method.

We copy the entire paragraph here for context:

"Among the main challenges in using satellite-derived AOD for this application are:

(1) The low temporal frequency of measurements from polar-orbiting instruments (i.e., at most, about once daily for MODIS, and between two and nine days for MISR, depending on latitude) compared to diurnally varying pollution levels in many settings

(2) Inaccuracies introduced in satellite aerosol retrieval algorithms by uncertain aerosol and surface optical properties

(3) The relatively coarse retrieval-product spatial resolution and aerosol species discrimination

(4) Inability to retrieve aerosol in the presence of cloud cover, and possible sub-pixel cloud contamination elsewhere (Duncan et al., 2014; Martonchik et al., 2009).

(5) The relationship between satellite-derived AOD and PM2.5 is not straightforward. AOD is the integral of atmospheric *optical* extinction from the surface to the top of the atmosphere under ambient temperature and humidity conditions, whereas PM2.5 is the near-surface aerosol *mass* concentration of dry particles with diameters < 2.5  $\mu$ m. The relationship depends upon the aerosol vertical distribution, hygroscopic growth factor, mass extinction efficiency, and ambient atmospheric relative humidity profile (Gupta et al., 2006). The relationship is also time dependent and can vary across typical satellite grid-cells (Engel-Cox et al., 2004; Hu, 2009; Lee et al., 2011)."

Line 98: The reviewer notes to detail the full acronym SPARTAN Thank you, we have reported the full name "To respond to this challenge, the **Surface PARTiculate mAtter Network (SPARTAN)** network"

Line 99: The reviewer notes: "also 7-SEAS NASA mission aims to setup permanent and mobile stations in wild and difficult accessible regions to monitor aerosols as shown in https://www.atmos-chem-phys.net/16/14057/2016/"

Thank you for this information! We see that the measurements for this mission are made by a cruise ship. Although we think such maritime measurements are exceedingly important, we see the most important use of our methodology for measuring  $PM_{2.5}$  concentrations over land, where people live.

Line 140: The reviewer notes that our characterizing of the OPC-N2 as unique is not fully true because many other sensors exist.

Thank you for this note. Although many other low-cost sensors exist, the OPC-N2 is the only low-cost sensor (< USD \$500) that has been shown to provide size resolved information- within a specific diameter range with any kind of accuracy. We note this in the text.

Line 261: The reviewer notes that our use of monthly effective fraction of each MISR component AOD to scale the more frequent MAIAC AODs is wrong. The reviewer again points this out for line 285

Please see our previous response

The manuscript with the highlighted changes is presented on the next page

**Combining low-cost, surface-based aerosol monitors with sizeresolved satellite data for air quality applications**

Priyanka deSouza1†, Ralph A Kahn2, James A Limbacher2, Eloise A. Marais3\*, Fábio Duarte1,4, Carlo Ratti1

[revised manuscript text omitted]

Diner, D.J., Boland, S.W., Brauer, M., Bruegge, C., Burke, K.A., Chipman, R., Di Girolamo, L., Garay, M.J., Hasheminassab, S., Hyer, E. and Jerrett, M., 2018. Advances in multiangle satellite remote sensing of speciated airborne particulate matter and association with adverse health effects: from MISR to MAIA. *Journal of Applied Remote Sensing*, *12*(4), p.042603.

Duncan, B.N., Prados, A.I., Lamsal, L.N., Liu, Y., Streets, D.G., Gupta, P., Hilsenrath, E., Kahn, R.A., Nielsen, J.E., Beyersdorf, A.J., Burton, S.P., Fiore, A.M., Fishman, J., Henze, D.K., Hostetler, C.A., Krotkov, N.A., Lee, P., Lin, M., Pawson, S., Pfister, G., Pickering, K.E., Pierce, R.B., Yoshida, Y., Ziemba, L.D., 2014. Satellite data of atmospheric pollution for U.S. air quality applications: Examples of applications, summary of data end-user resources, answers to FAQs, and common mistakes to avoid. Atmospheric Environment 94, 647–662. https://doi.org/10.1016/j.atmosenv.2014.05.061

EC-JRC/PBL–European Commission, 2011. Joint Research Centre/Netherlands Environmental Assessment Agency PBL: Emission Database for Global Atmospheric Research (EDGAR), Release Version 4.2.

Engel-Cox, J.A., Holloman, C.H., Coutant, B.W., Hoff, R.M., 2004. Qualitative and quantitative evaluation of MODIS satellite sensor data for regional and urban scale air quality. Atmospheric Environment 38, 2495–2509. https://doi.org/10.1016/j.atmosenv.2004.01.039

Friberg, M.D., Kahn, R.A., Limbacher, J.A., Appel, K.W., Mulholland, J.A., 2018. Constraining chemical transport PM2.5 modeling outputs using surface monitor measurements and satellite retrievals: application over the San Joaquin Valley. Atmospheric Chemistry & Physics 18, 12891–12913. https://doi.org/10.5194/acp-18-12891-2018

Gupta, P., Christopher, S.A., Wang, J., Gehrig, R., Lee, Y., Kumar, N., 2006. Satellite remote sensing of particulate matter and air quality assessment over global cities. Atmospheric Environment 40, 5880–5892. https://doi.org/10.1016/j.atmosenv.2006.03.016 Hagan, D. H. and Kroll, J. H.: Assessing the accuracy of low-cost optical particle sensors using a physics-based approach, Atmos. Meas. Tech. Discuss., https://doi.org/10.5194/amt-2020-188, in review, 2020.

Hu, Z., 2009. Spatial analysis of MODIS aerosol optical depth, PM2.5, and chronic coronary heart disease. International Journal of Health Geographics 8, 27. https://doi.org/10.1186/1476-072X-8-27

Kahn, R., Banerjee, P., McDonald, D., 2001. Sensitivity of multiangle imaging to natural mixtures of aerosols over ocean. Journal of Geophysical Research: Atmospheres 106, 18219–18238. https://doi.org/10.1029/2000JD900497

Kahn, R.A., Gaitley, B.J., 2015. An analysis of global aerosol type as retrieved by MISR. Journal of Geophysical Research: Atmospheres 120, 4248–4281. https://doi.org/10.1002/2015JD023322

Kahn, R.A., Gaitley, B.J., Garay, M.J., Diner, D.J., Eck, T.F., Smirnov, A., Holben, B.N., 2010. Multiangle Imaging SpectroRadiometer global aerosol product assessment by comparison with the Aerosol Robotic Network. Journal of Geophysical Research: Atmospheres 115. https://doi.org/10.1029/2010JD014601

Kumar, P., Morawska, L., Martani, C., Biskos, G., Neophytou, M., Di Sabatino, S., Bell, M., Norford, L., Britter, R., 2015. The rise of low-cost sensing for managing air pollution in cities. Environment International 75, 199–205. https://doi.org/10.1016/j.envint.2014.11.019

Lee, H.J., Liu, Y., Coull, B.A., Schwartz, J., Koutrakis, P., 2011. A novel calibration approach of MODIS AOD data to predict PM 2.5 concentrations. https://doi.org/10.5194/acp-11-7991-2011

Levy, R.C., Mattoo, S., Munchak, L.A., Remer, L.A., Sayer, A.M., Patadia, F., Hsu, N.C., 2013. The Collection 6 MODIS aerosol products over land and ocean. Atmospheric Measurement Techniques 6, 2989–3034. https://doi.org/10.5194/amt-6-2989-2013

Lewis, A., Edwards, P., 2016. Validate personal air-pollution sensors. Nature 535, 29–31. https://doi.org/10.1038/535029a

Limbacher, J.A., Kahn, R.A., 2017. Updated MISR dark water research aerosol retrieval algorithm – Part 1: Coupled 1.1 km ocean surface chlorophyll *a* retrievals with empirical calibration corrections. Atmospheric Measurement Techniques 10, 1539–1555. https://doi.org/10.5194/amt-10-1539-2017

Limbacher, J.A., Kahn, R.A., 2015. MISR empirical stray light corrections in high-contrast scenes. Atmospheric Measurement Techniques 8, 2927–2943. https://doi.org/10.5194/amt-8-2927-2015

Limbacher, J.A., Kahn, R.A., 2014. MISR Research Aerosol Algorithm: refinements for dark water retrievals. Atmospheric Measurement Techniques Discussions 7, 7837–7882. https://doi.org/10.5194/amtd-7-7837-2014

Limbacher, J., Kahn, R.A., 2019. Updated MISR Over-Water Research Aerosol Retrieval Algorithm - Part 2: A Multi-Angle Aerosol Retrieval Algorithm for Shallow, Turbid, Oligotrophic, and Eutrophic Waters. Atmospheric Measurement Techniques 675–689. https://doi.org/10.5194/amt-12-675-2019, http://dx.doi.org/10.5194/amt-12-675-2019

[revised manuscript text omitted]